# SubTrack your Grad: Gradient Subspace Tracking for Memory-Efficient LLM Training and Fine-Tuning

## Abstract

Training and fine-tuning Large Language Models (LLMs) demand significant computational resources and time due to their large model sizes and optimizer states. To mitigate these challenges and improve accessibility, several memory-efficient methods have been developed. Methods such as Low-Rank Adaptation (LoRA) optimize model weights within a low-rank subspace, while Gradient Low-Rank Projection (GaLore) projects gradients into a lower-dimensional space to decrease memory footprint. In this paper, we propose Gradient Subspace Tracking (SubTrack-Grad), a method that confines optimization to a compact core subspace of the gradient matrices and dynamically tracks its changes using the geometry of Grassmannian manifolds. SubTrack-Grad efficiently updates its subspace estimation by leveraging estimation errors and previously identified subspaces. Specifically, SubTrack-Grad **reduces wall-time by up to 20.57% on GLUE tasks (15% average reduction)** and **up to 65% on SuperGLUE tasks (22% average reduction).** Notably, for a 3B parameter model, GaLore incurred a substantial 157% increase in wall-time compared to full-rank training, whereas SubTrack-Grad exhibited only a 31% increase—**representing a 49% reduction in wall-time compared to GaLore.** Additionally, the memory required for storing optimizer states in SubTrack-Grad is equivalent to GaLore's, and it exhibits only a minimal increase in peak memory consumption.

## 1 Introduction

Large Language Models (LLMs) have achieved state-of-the-art performance across various tasks and are rapidly growing in popularity. However, their training and fine-tuning demand substantial resources, such as hardware and time, making them impractical for many applications and contributing to a larger carbon footprint (Zhao et al., 2024; Jaiswal et al., 2024; Muhamed et al., 2024; Miles et al., 2024; Modoranu et al., 2024; Hao et al., 2024; Li et al., 2024). As a result, there is an acute need to develop memory- and time-efficient methods to democratize their use and mitigate environmental impact. Various techniques have been proposed to reduce memory usage, such as gradient checkpointing (Chen et al., 2016) and memory offloading (Rajbhandari et al., 2020). In this context, several Parameter-Efficient Fine-Tuning (PEFT) approaches aim to reduce memory usage by optimizing a subset of model parameters or operating within a lower-dimensional space (Dettmers et al., 2024; Yaras et al., 2024; Lialin et al., 2023; Renduchintala et al., 2024; Xia et al., 2024; Miles et al., 2024; Hu et al., 2021). Notably, the well-known method LoRA (Hu et al., 2021) decomposes weight matrices into two low-rank trainable matrices, optimizing network parameters within a small subspace, which significantly reduces the memory footprint.

Memory requirements extend beyond trainable parameters, with a significant portion consumed by optimizers for storing element-wise states and parameters (Zhao et al., 2024). To address this, recent efforts have focused on reducing the memory footprint of optimizer parameters (Li et al., 2023; Anil et al., 2019; Lv et al., 2024; Dettmers et al., 2022; Zhang et al., 2024; Modoranu et al., 2024; Zhao et al., 2024; Muhamed et al., 2024). GaLore (Zhao et al., 2024) reduces the memory usage of optimizers by projecting gradient matrices into a low-rank subspace and tracking changes via periodic Singular Value Decomposition (SVD) to obtain a rank-$r$ approximation. However, this approach faces several challenges. First, SVD is computationally expensive, and if the gradient does

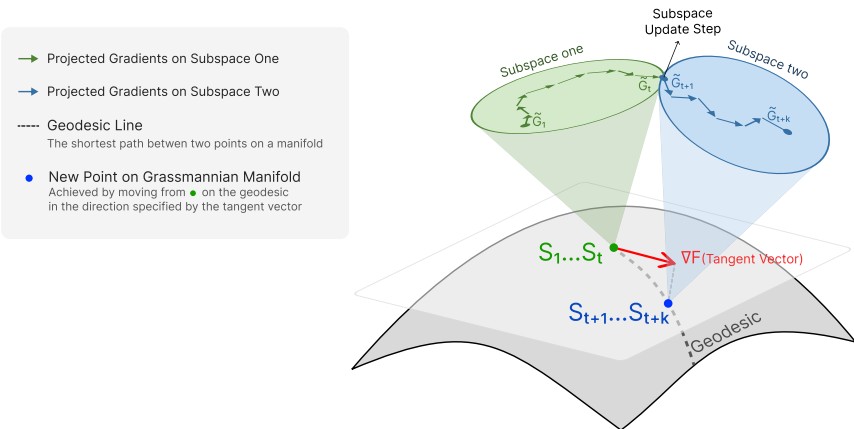

Figure 1: Visualization of the SubTrack-Grad method: Between subspace update steps, gradients are projected onto a fixed subspace. The tangent vector $\nabla F$ is computed via the derivative of a loss function that measures the subspace estimation error between updates. The subspace is then updated by moving along the corresponding geodesic, determined by $\nabla F$, on the Grassmannian manifold to minimize the measured error.

not evolve within a nearly constant subspace, GaLore must increase the frequency of SVD operations, significantly increasing the amount of computation. This is problematic because not all layers' gradients converge to a stable subspace early in training (Jaiswal et al., 2024). Moreover, applying SVD to a single gradient matrix is susceptible to data noise (Vaswani et al., 2018), and GaLore does not leverage 1) information from the orthogonal space as feedback to adjust the subspace (Modoranu et al., 2024) or 2) previously computed subspaces to incorporate past knowledge, which could help mitigate these effects and improve convergence speed.

To address these challenges, we propose Gradient Subspace Tracking (SubTrack-Grad), a Grassmannian-based subspace tracking method that efficiently updates the subspace using rank-1 updates. SubTrack-Grad leverages information from the orthogonal complement to improve subspace estimation through simple linear algebra operations, which are computationally more efficient than GaLore as they avoid periodic SVD on the main gradient matrices. Furthermore, SubTrack-Grad dynamically adapts to changes in the gradient subspace, reducing abrupt shifts in subspace updates for faster convergence. Our main contributions are as follows:

- We introduce SubTrack-Grad, a computationally and memory-efficient method that projects gradients onto a core subspace and dynamically adjusts this subspace using Grassmannian manifold geometry, leveraging estimation errors as a signal for adjustment.

- We demonstrate that SubTrack-Grad achieves performance comparable to or better than GaLore, with a significant reduction in runtime and minimal memory overhead.

- We show that tracking the gradient subspace helps reduce abrupt changes in the optimization process, thereby accelerating convergence.

- We prove that our method aligns with GaLore's convergence guarantees while enabling more frequent subspace updates by exercising greater control over subspace adjustments through the use of prior knowledge and errors from the orthogonal space.

An overview of the proposed method is provided in Figure 1.

## 2 RELATED WORKS

Several works aim to improve the efficiency of training and fine-tuning LLMs, addressing a growing demand as their popularity rapidly increases. LoRA (Hu et al., 2021), a widely recognized method for reducing the number of trainable parameters, projects model weights into a lower-dimensional space, resulting in two trainable low-rank matrices. This approach optimizes the matrices and significantly reduces memory requirements for fine-tuning large models. Dettmers et al. (2024) builds

on LoRA by employing quantization techniques and paged optimizers to further reduce memory usage. Additionally, Yaras et al. (2024) introduces Deep LoRA, which uses deep matrix factorization for low-rank optimization, addressing overfitting issues and reducing the need for precise tuning of the rank parameter. Several other works have also extended LoRA to enhance the efficiency of training and fine-tuning large models (Lialin et al., 2023; Renduchintala et al., 2024; Xia et al., 2024; Pan et al., 2024). Miles et al. (2024) proposes compressing intermediate activation vectors and reconstructing them during backpropagation to enhance memory efficiency. Additionally, Hao et al. (2024) demonstrates that full-parameter fine-tuning is feasible by using random projections on the gradient matrix, showing that LoRA essentially performs a down-projection of the gradient. BAdam (Luo et al., 2024) leverages the block coordinate descent framework to achieve low memory consumption while maintaining optimization capabilities comparable to Adam.

Several approaches aim to reduce memory consumption in optimizers, as optimizers like Adam (Kingma & Ba, 2017) account for a significant portion of memory usage due to their storage of element-wise states to improve the optimization process (Li et al., 2023; Anil et al., 2019; Lv et al., 2024; Dettmers et al., 2022). MicroAdam (Modoranu et al., 2024) tackles this issue by compressing the gradient space for optimization and utilizing the resulting compression error through feedback loops to improve the optimization process. Adam-mini (Zhang et al., 2024) partitions model parameters into blocks, assigning a single learning rate to each block. This design achieves a significant reduction in memory usage while maintaining performance. Gur-Ari et al. (2018) suggests that a substantial portion of gradients lies within a small, largely consistent subspace, a finding also reported by other studies, including Schneider et al. (2024); Yaras et al. (2023). GaLore (Zhao et al., 2024) leverages this property to reduce the memory requirements of optimizers by projecting gradients into a lower-dimensional subspace and then projecting them back for complete parameter tuning. This approach has been effectively integrated with other methods to further reduce memory usage during training and fine-tuning of LLMs (Li et al., 2024). However, not all layers' gradients evolve within a stable low-rank subspace. Jaiswal et al. (2024) identifies layers with constantly changing gradients where low-rank projection may be inefficient for tuning. By analyzing the distribution of singular values across different layers, they select those that evolve within a small subspace for fine-tuning while freezing the remaining layers. Gradient Structured Sparsification (Grass) (Muhamed et al., 2024) further reduces memory usage by applying sparse projection matrices to the gradient, transforming the gradient matrix into a sparse vector space. This approach leverages sparse representations to significantly decrease the memory footprint. In Ramesh et al. (2024) the authors propose an approach that dynamically selects and updates a small subset of parameters, leading to faster and more memory-efficient training without altering the model's structure.

When working with high-dimensional data, a common approach is to project the data into a lower-dimensional space, and many studies focus on tracking these subspaces as they evolve over time. Balzano et al. (2011) introduces an incremental method for updating subspaces on the Grassmannian manifold when data is partially observed. Zhang & Balzano (2016) and Kasai (2017) address the challenges posed by noise in data streams and evolving environments, proposing methods to handle such noise. Additionally, Blocker et al. (2023) presents a method for time-varying data based on geodesics in Grassmannian space to track changes and update the subspace effectively.

## 3 SUBTRACK-GRAD: TRACKING THE GRADIENT SUBSPACE

Since gradients typically evolve within a small subspace, compressing this space can significantly reduce the memory footprint of the optimizer. As demonstrated in Zhao et al. (2024) whenever the gradient takes the general form

$$G = \sum_i A_i + \sum_i B_i W C_i \tag{1}$$

where $i$ denotes the batch index, and $B_i$ and $C_i$ are positive semi-definite (PSD) matrices, this gradient can be projected onto a small subspace that remains nearly stable while ensuring that the optimization process continues to converge, as discussed in Section 4 and Appendix A. However, the gradient's subspace is not always stable, making it crucial to track its changes for effective optimization. GaLore (Zhao et al., 2024) addresses this by periodically performing SVD on gradient matrices, while keeping the update frequency low to align with the assumption of a stable subspace. This approach poses several challenges: **1)** not all gradients converge to a stable subspace within a few iterations, **2)** SVD is computationally expensive, and increasing the frequency of updates to

---

**Algorithm 1** SubTrack-Grad

---

**Require:** Sequence of $m \times n$ gradients $G_t$ with $m \leq n$ (w.l.o.g.), step-size $\eta$, rank $r$, subspace update steps k

    **Initialize Subspace via SVD Decomposition:**

    $P_0 \leftarrow U[:, :r]$, where $U, S, V \leftarrow \mathrm{SVD}(G_0)$

    $S_0 \leftarrow P_0$                                                       {The initial subspace}

    **for** $t = 1, \ldots, T$ **do**

      **if** $t \mod k = 0$ **then**

        **Update subspace:**

        $G_{lr} = \arg\min_A \|(S_{t-1}A - G_t)\|^2$           {Solving the least square problem}

        $R = G_t - S_{t-1}G_{lr}$                           {Computing the residual}

        $\nabla F = -2RG_{lr}^\top \approx \widehat{U}_F \widehat{\Sigma}_F \widehat{V}_F^\top$    {Computing the rank-1 estimation of tangent vector}

        $S_t = (S_{t-1}\widehat{V}_F \quad \widehat{U}_F)\begin{pmatrix}\cos\widehat{\Sigma}_F\eta \\ \sin\widehat{\Sigma}_F\eta\end{pmatrix}\widehat{V}_F^\top + S_{t-1}(I - \widehat{V}_F\widehat{V}_F^\top)$   {Updating the subspace}

      **else**

        **Keep using previous subspace:** $S_t = S_{t-1}$

      **Return final projected gradient to the optimizer:** $S_t^\top G_t$

---

capture changes significantly raises both runtime and environmental costs, and **3)** increasing the update frequency contradicts the assumption of a stable subspace, as SVD is sensitive to noise and does not account for the previously computed subspace or estimation error to regulate the extent of change applied.

We propose Gradient Subspace Tracking, or SubTrack-Grad, a computationally efficient method for tracking gradient subspaces. SubTrack-Grad utilizes both the orthogonal space and the previously computed subspace to update the core subspace. Also, ass illustrated in 4, increasing the update frequency does not significantly impact runtime. This approach effectively controls the amount of change in the subspace, enabling more frequent updates without compromising the stability assumption. The subspace is initialized using SVD as follows:

$$G_0 = USV^\top \approx \sum_{i=1}^{r} s_i u_i v_i^\top, \quad P_0 = [u_1, u_2, ..., u_r], \quad Q_0 = [v_1, v_2, ..., v_r]. \tag{2}$$

Here, $G_0$ is an $m \times n$ gradient matrix at step 0, and $U$, $S$, and $V$ are its SVD components, with $r$ denoting the specified rank. At each optimization step, the gradients are projected onto the subspace of the left singular vectors if $m \leq n$, or onto the right singular vectors otherwise, thereby optimizing memory usage (Zhao et al., 2024). The optimization is performed within this subspace, after which the gradient is projected back to allow full parameter tuning. For simplicity, we assume $m \leq n$ without loss of generality, implying that $S_0 = P_0$, an $m \times r$ orthonormal matrix whose columns span the underlying subspace.

At each iteration of pre-training or fine-tuning, the matrix $S_t$, representing the subspace at step $t$, projects the gradient matrix $G_t$ onto the subspace by $\widetilde{G}_t = S_t^\top G_t$, where $\widetilde{G}_t$ will be a reduced $r \times n$ matrix representing the projection of the original gradient onto a rank-$r$ subspace. The optimizer then performs optimization within this low-rank space, which substantially reduces the number of state parameters and thus the memory usage. The optimizer outputs $\widetilde{G}_t^O$, which is then projected back to the original space by $\widehat{G}_t = S_t \rho_t \widetilde{G}_t^O$, where $\rho_t$ is a scalar representing the entry-wise regularizer used in the optimizer, to be passed to the network.

As previously discussed, the gradient does not always evolve within a stable low-rank subspace; hence, $S_t$, the orthonormal matrix spanning the core subspace, must be appropriately updated. In SubTrack-Grad, we propose updating the subspace by moving along Grassmannian geodesics. This approach leverages the previously computed subspace and the estimation error from earlier steps to minimize abrupt changes and noise effects, as illustrated in Figure 2.

The underlying subspace is updated after a fixed subspace update interval of $k$ steps. Let $T_i$ denote the $i$-th subspace update step for $i \in \{1, 2, 3, \ldots\}$. To mitigate the impact of noise, we can compute an accumulated gradient by averaging gradients across consecutive subspace update steps, as

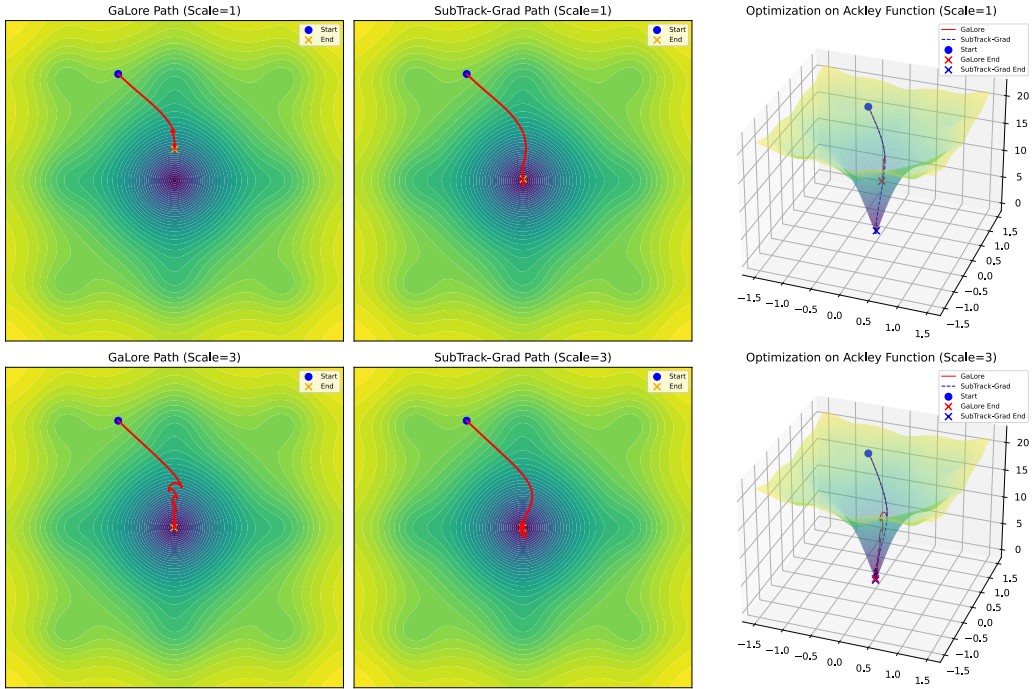

Figure 2: Comparison of the Adam Optimizer Combined with GaLore and SubTrack-Grad on the Ackley Function. Both optimizers took 100 steps, with the subspace update interval set to 10. As demonstrated, when the scale factor is equal to 1, GaLore could not reach the global minimum of the function within 100 iterations due to more frequent jumps compared to SubTrack-Grad. For a scale factor of 3, the length of jumps increases further in Adam combined with GaLore. The 3D plots provide a clearer visualization of the dynamics of the Ackley function and the optimization paths followed by each method in a single plot.

described in equation 3. However, our experiments, summarized in Table 1, indicate that in most cases, using only the latest gradient achieves comparable results while reducing memory usage as shown in Table 4. Furthermore, as evidenced in Tables 1, 2, and 3, this approach either outperforms GaLore or delivers similar performance, even without relying on the accumulated gradient.

$$G_{acc} = \frac{1}{T_n - T_{n-1}} \sum_{t=T_{n-1}}^{T_n} G_t \tag{3}$$

We frame the problem of identifying the subspace as selecting the appropriate element from the Grassmannian, the set of all $d$-dimensional subspaces within an $n$-dimensional vector space (Bendokat et al., 2024). Our objective is to minimize the Euclidean distance between the current subspace and the observed gradient at each update step. The cost function is defined as:

$$F(S_t) = \min_A \|S_t A - G_t\|_F^2, \tag{4}$$

where $A$ is the solution to the least squares problem. The derivative of this function with respect to $S_t$ is given in equation 5, and $R = G_t - S_t A$ lies in the orthogonal complement of $S_t$. To update the subspace in the appropriate direction, we compute the tangent vector $\nabla F$ on the Grassmannian manifold, as shown in equation 6 based on Edelman et al. (1998), where the second equality holds because $R$ is orthogonal to $S_t S_t^\top$.

$$\frac{\partial F}{\partial S_t} = 2(S_t A - G_t) A^\top = -2RA^\top \tag{5}$$

$$\nabla F = (I - S_t S_t^\top)\frac{\partial F}{\partial S_t} = \frac{\partial F}{\partial S_t} = -2RA^\top \approx \widehat{U}_F \widehat{\Sigma}_F \widehat{V}_F^\top \tag{6}$$

The tangent vector $\nabla F$ provides the direction for adjusting the subspace by accounting for the error lying in the orthogonal complement. However, to minimize changes to the subspace, SubTrack-Grad first computes a rank-1 approximation of $\nabla F$, determined by its largest singular value and the

corresponding singular vectors obtained from its SVD, represented as $\widehat{U}_F\widehat{\Sigma}_F\widehat{V}_F^\top$ in the final equality of equation 6. This approximation is then used to update the subspace.

As shown by Edelman et al. (1998); Bendokat et al. (2024), from which we have included the necessary definitions and theorem in section 4, we can move along the Grassmannian geodesic in the direction of the computed tangent vector, taking a step of size $\eta$, as presented in equation 7.

$$S_{t+1}(\eta) = (S_t\widehat{V}_F \quad \widehat{U}_F) \begin{pmatrix} \cos\widehat{\Sigma}_F\eta \\ \sin\widehat{\Sigma}_F\eta \end{pmatrix} \widehat{V}_F^\top + S_t(I - \widehat{V}_F\widehat{V}_F^\top) \tag{7}$$

This update step preserves the orthonormality of $S_{t+1}$, ensuring it remains on the Grassmannian manifold. The last term in equation 6, which is required when using thin-SVD instead of compact-SVD, projects the previous subspace onto the orthogonal complement of $\widehat{V}_F$. This ensures that the portion of the subspace of $S_t$ not updated in this step is still included. By leveraging the geometry of the Grassmannian manifold, SubTrack-Grad effectively tracks the underlying subspace of the gradient space. The pseudo-code for this method is provided in Algorithm 1.

## 4 THEORETICAL ANALYSIS

In this section, we analyze the convergence of SubTrack-Grad using theoretical analysis. To begin, using SubTrack-Grad, the update rule for the weights of the networks is as follows:

$$W_t = W_0 + \sum_{t'=0}^{t'=t-1} \widehat{G}_{t'} \tag{8}$$

As previously mentioned, we use left projection if $m \leq n$, where $m$ and $n$ are the dimensions of the original gradient matrix, and right projection otherwise. Thus, $\widehat{G}_{t'}$ can be computed as shown in equation 9.

$$\widehat{G}_{t'} = \begin{cases} S_{t'}\rho_{t'}(S_{t'}^\top G_{t'}), & \text{if } m \leq n \\ \rho_{t'}(G_{t'}S_{t'})S_{t'}^\top, & \text{otherwise} \end{cases} \tag{9}$$

Here, $S_{t'}$ is the projection matrix that projects the gradient onto the subspace, and $\rho_{t'}$ is a scalar representing the entry-wise regularizer used in the optimizer. If we use the full projection, then $\widehat{G}_{t'}$ will be computed as shown below:

$$\widehat{G}_{t'} = S_{t'}^l\rho_{t'}(S_{t'}^{l\top} G_{t'}S_{t'}^r)S_{t'}^{r\top} \tag{10}$$

where $S_{t'}^l$ and $S_{t'}^r$ are the rank-$r$ left and right projection matrices.

**Definition 4.1 ( L-continuity)** *A function $f(X)$ has Lipschitz-continuity (L-continuity) if for any $X_1$ and $X_2$, $\|f(X_2) - f(X_1)\|_F \leq L\|X_2 - X_1\|_F$*

**Theorem 4.1 (Convergence of SubTrack-Grad)** *Suppose gradient has the following form (also equation 1) with functions $A_i$, $B_i$, and $C_i$ being L-continuous as per **Def. B.1** with constants $L_A$, $L_B$, and $L_C$ w.r.t. weight matrix $W_t$; and $\|W_t\|_F \leq M$; where $W_t$ denotes the weight matrix at step $t$, and $M$ is a scalar value,*

$$G = \sum_i A_i + \sum_i B_i W C_i.$$

*Now, define $\widehat{B}_{i,t} = (S_{i,t}^l)^\top B_i(W_t)S_{i,t}^l$ and $\widehat{C}_{i,t} = (S_{i,t}^r)^\top C_i(W_t)S_{i,t}^r$, where $S_{i,t}^l$ and $S_{i,t}^r$ are the rank-r left and right projection matrices; $B_i(W_t)$ and $C_i(W_t)$ denote the dependence of $B_i$ and $C_i$ on the weight matrices $W_t$. Further letting $P_t = S_t^{l\top} G_t S_t^r$, and $\kappa_t = \frac{1}{N}\sum_i \lambda_{min}(\widehat{B}_{i,t})\lambda_{min}(\widehat{C}_{i,t})$, where $\lambda_{min}(\cdot)$ denotes the minimum eigenvalue over each batch, and N representing the number of samples in a batch. Assuming that the projection matrices remain constant during the training. Then for learning-rate $\mu$ and $min(\kappa_t) > (L_A + 2L_BL_CM^2)$, the SubTrack-Grad, with $\rho_t \equiv 1$ (the element-wise regularizer of the optimizer) satisfies:*

$$\|P_t\|_F \leq [1 - \mu(\kappa_{t-1} - L_A - 2L_BL_CM^2)]\|P_{t-1}\|_F.$$

*That is, $P_t \to 0$ and SubTrack-Grad converges.*

The proof of Theorem 4.1 is provided in Appendix A, based on Zhao et al. (2024). Note that while both GaLore and SubTrack-Grad assume that the projection matrices remain unchanged for the proof of convergence, GaLore must limit the number of updates to ensure convergence, as each update can potentially change the subspace entirely. In contrast, SubTrack-Grad leverages only rank-1 updates to the subspace, preventing drastic changes with each update. While a deeper analysis of slowly changing subspaces and their impact on convergence remains an open problem, in practice, this allows SubTrack-Grad to perform more updates than GaLore.

The Grassmannian update rule presented in equation 7 is a direct application of Grassmann geometry (Edelman et al., 1998; Bendokat et al., 2024).

**Definition 4.2 (Exponential Map)** *The exponential map* $\exp_p : T_pM \to M$ *on a Riemannian manifold $M$ is a mapping that assigns to each tangent vector $\Delta \in T_pM$ the point $\gamma(1) \in M$, where $T_pM$ is the tangent space of $M$ at $p$, and $\gamma$ is the unique geodesic originating at $p$ with initial velocity $\Delta$. This map establishes a relationship between geodesics and the Riemannian exponential, such that $\gamma(t) = \exp_p(t\Delta)$ for $t \in \mathbb{R}$.*

**Definition 4.3 (Stiefel Manifold)** *The Stiefel manifold $St(n, p)$ parametrizes the set of all $n \times p$ matrices $U$, with orthonormal columns, each representing a rank-$p$ subspace of $\mathbb{R}^n$.*

**Definition 4.4 ( Grassmann Manifold)** *The Grassmannian manifold $Gr(n, p)$ parametrizes the set of all $p$-dimensional subspaces of $\mathbb{R}^n$. Each point can be represented by a projection matrix $P = UU^T$, where $U \in St(n, p)$.*

**Theorem 4.2 (Grassmann Exponential)** *Let $P = UU^T \in Gr(n, p)$ be a point on the Grassmannian manifold, where $U \in St(n, p)$ is the orthonormal basis of the corresponding subspace. Consider a tangent vector $\Delta \in T_PGr(n, p)$, and let $\Delta_U^{hor}$ denote the horizontal lift of $\Delta$ to the horizontal space at $U$ in the Stiefel manifold $St(n, p)$. Suppose the thin SVD of $\Delta_U^{hor}$ is given by $\Delta_U^{hor} = \hat{Q}\Sigma V^T$, where $\hat{Q} \in St(n, r)$, $\Sigma = diag(\sigma_1, \ldots, \sigma_r)$ contains the nonzero singular values of $\Delta_U^{hor}$ with $r = \min(p, n - p)$, and $V \in St(p, r)$. The Grassmann exponential map, representing the geodesic emanating from $P$ in the direction $\Delta$, is given by:*

$$Exp_P^{Gr}(t\Delta) = \left[ UV\cos(t\Sigma)V^T + \hat{Q}\sin(t\Sigma)V^T + UV_\perp V_\perp^T \right],$$

*where $V_\perp \in \mathbb{R}^{p\times(p-r)}$ is any orthogonal complement of $V$.*

The proof of Theorem 4.2 can be found in Appendix B. Leveraging this theorem, and incorporating our notation in section 3, one can easily verify that the subspace update rule is as follows:

$$S_{t+1}(\eta) = (S_t\widehat{V}_F \quad \widehat{U}_F) \begin{pmatrix} \cos\widehat{\Sigma}_F\eta \\ \sin\widehat{\Sigma}_F\eta \end{pmatrix} \widehat{V}_F^\top + S_t(I - \widehat{V}_F\widehat{V}_F^\top)$$

This update rule generally converges to a stable subspace if the step size $\eta$ decreases over time (Balzano et al., 2011). However, a decreasing step size can impair the ability to accurately track and adapt to subspace changes. Consequently, SubTrack-Grad uses a constant step size during training and fine-tuning to effectively adjust subspaces. This approach does not hinder convergence, as proved in Theorem 4.1, which guarantees convergence as long as changes are controlled to maintain the stable subspace assumption.

## 5 EXPERIMENTS AND RESULTS

To ensure a fair comparison between SubTrack-Grad and baselines, we conducted fine-tuning and pre-training experiments across various architectures and datasets; to measure performance, wall-time, and memory consumption, while all shared hyperparameters remaine consistent.

**Fine-Tuning Experiments.** RoBERTa-Base was fine-tuned on GLUE tasks, while RoBERTa-Large was used for fine-tuning on SuperGLUE datasets. For performance evaluation, the models were fine-tuned on each task for 30 epochs, with the results for GLUE tasks presented in Table 1, and SuperGLUE results shown in Table 2. Table 1 compares the performance between using accumulated

Table 1: Evaluating the performance of SubTrack-Grad and other baselines when fine-tuning RoBERTa-Base on GLUE tasks for different ranks $r$. All hyperparameters are the same for each rank. The performance is measured via Accuracy for QNLI, MNLI, SST-2, and RTE tasks, F1 for QQP and MRPC, Pearson Correlation for STS-B, and Matthews Correlation for COLA, after fine-tuning for 30 epochs.

| | COLA | STS-B | MRPC | RTE | SST-2 | MNLI | QNLI | QQP | Avg |
|---|---|---|---|---|---|---|---|---|---|
| **Full-Rank** | 62.57 | 91.03 | 91.32 | 77.98 | 94.27 | 87.83 | 92.71 | 89.21 | 85.86 |
| **GaLore, r=4** (Zhao et al., 2024) | 60.34 | 90.58 | 92.58 | 76.53 | 94.27 | 87.12 | 92.20 | 87.86 | 85.18 |
| **SubTrack(Acc)-Grad, r=4** (Ours) | 61.32 | 90.64 | 92.66 | 77.98 | 94.15 | 86.85 | 91.85 | 87.50 | **85.37** |
| **SubTrack(Last)-Grad, r=4** (Ours) | 61.07 | 90.63 | 92.83 | 76.89 | 93.81 | 87.20 | 91.73 | 86.73 | 85.11 |
| **GaLore, r=8** (Zhao et al., 2024) | 58.54 | 90.61 | 91.30 | 74.37 | 94.50 | 87.34 | 92.71 | 87.99 | 84.67 |
| **SubTrack(Acc)-Grad, r=8** (Ours) | 58.54 | 90.87 | 91.43 | 76.53 | 94.27 | 87.09 | 92.49 | 87.57 | 84.85 |
| **SubTrack(Last)-Grad, r=8** (Ours) | 58.03 | 90.76 | 91.81 | 77.62 | 94.38 | 87.15 | 92.31 | 87.26 | **84.91** |

Table 2: Evaluating the performance of SubTrack-Grad and other baselines when fine-tuning RoBERTa-Large on SuperGLUE tasks with rank 8. All hyperparameters are consistent across different methods. The performance is measured via Accuracy for COPA, WIC, WSC, BoolQ, and $AX_g$ tasks, F1 for CB, and Matthews Correlation for $AX_b$, after fine-tuning for 30 epochs.

| | BoolQ | CB | COPA | WIC | WSC | $AX_b$ | $AX_g$ | Avg |
|---|---|---|---|---|---|---|---|---|
| **Full-Rank** | 85.29 | 91.76 | 48.00 | 71.94 | 63.46 | 57.68 | 100 | 74.02 |
| **GaLore** (Zhao et al., 2024) | 86.42 | 88.97 | 48.00 | 71.32 | 63.46 | 45.92 | 98.15 | 71.75 |
| **SubTrack-Grad** (Ours) | 85.81 | 91.76 | 53.00 | 71.47 | 63.46 | 47.93 | 100 | **73.35** |

Table 3: Evaluation loss comparison for pre-training different Llama-based architectures on C4 dataset after 10K iterations.

| | 60M rank=128 | 130M rank=256 | 350M rank=256 | 1B rank=512 | 3B rank=512 | Avg - |
|---|---|---|---|---|---|---|
| **Full-Rank** | 6.27 | 6.60 | 6.40 | 6.06 | 6.42 | 6.35 |
| **GaLore (Zhao et al., 2024)** | 3.65 | 3.44 | 3.52 | 5.59 | 5.90 | **4.42** |
| **SubTrack-Grad (Ours)** | 3.72 | 3.48 | 3.53 | 6.03 | 6.00 | 4.55 |

gradients and the last gradient matrix. In other experiments, we report results without gradient accumulation, increasing memory efficiency while maintaining performance. Figures 3b, 3e, 3c, and 3f compare wall-time and memory consumption during fine-tuning, with detailed reports provided in Appendix C. For wall-time comparisons, models were fine-tuned up to a number of iterations ensuring exactly five subspace updates, excluding evaluation steps, to provide an accurate runtime comparison. SubTrack-Grad demonstrates significant efficiency improvements, **reducing wall-time by up to 20.57% on GLUE tasks (15% average reduction)** and **up to 65% on SuperGLUE tasks (22% average reduction).** Despite limiting updates to rank-1 modifications of the previous subspace, Tables 1 and 2 shows that SubTrack-Grad achieves comparable or even superior performance compared to GaLore. Details of hyperparameters are provided in Appendix D.

**Pre-Training Experiments.** Different Llama-based architectures were pre-trained on C4 datasets. Each architecture was trained for 10K steps, and their evaluation losses are reported in Table 3. The diminishing increase in evaluation loss with larger model sizes highlights the importance of subspace tracking in mitigating abrupt changes, particularly when modeling more complex architectures using low-rank representations. For wall-time comparison, each architecture was trained for 2K iterations, with the subspace update interval set to 200, ensuring exactly 10 subspace updates. For memory

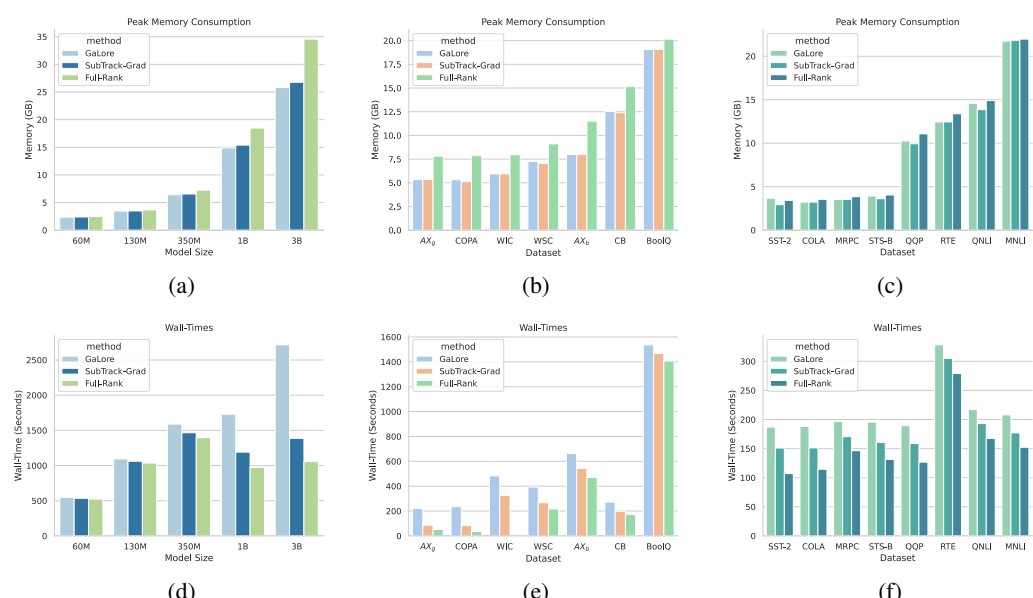

Figure 3: Comparing the peak memory consumption and wall-time on different architecture and datasets. (a) and (d) show the memory and wall-time measured on pretraining different Llama-architectures sizing from 60M to 3B. (b) and (e) show the memory and wall-time measured on fine-tuning RoBERTa-Large on SuperGLUE tasks. (c) and (f) show the memory and wall-time measured on fine-tuning RoBERTa-Base on GLUE tasks. As demonstrated, GaLore can lead to 2.5 times longer wall-time in training Llama with 3B parameters, while SubTrack-Grad significantly reduces this overhead, and shows minor additional memory requirements.

comparison, we used a batch size of 8, the largest size feasible on an NVIDIA A100 GPU with 40GB memory for the Llama model with 3B parameters. Figures 3a and 3d illustrate the memory and wall-time comparisons, with detailed reports provided in Appendix C. Notably, GaLore incurred a substantial 157% increase in wall-time compared to full-rank training for the model of size 3B, whereas SubTrack-Grad exhibited only a 31% increase—**representing a 49% reduction in wall-time compared to GaLore.** On average, SubTrack-Grad required **27% less training time** across these models, while exhibiting at most 3.7% memory overhead on the model with 3B parameters. Additional architectural details for the Llama-based models are provided in Appendix E.

**Time and Space Complexity.** Table 4 provides memory requirements of the optimizer states and the time complexity of the SVD operation considering an $m \times n$ gradient matrix with $m \leq n$. As discussed earlier, GaLore performs SVD on the gradient matrix to estimate the underlying subspace, whereas SubTrack-Grad performs SVD on the rank-$r$ tangent vector to update the subspace. Since SVD is the a computationally expensive operation, comparing its time complexity highlights why SubTrack-Grad is significantly more efficient than GaLore. As shown in Figure 3d, for pre-training a Llama model of size 3B, GaLore incurs over 157% wall-time overhead, compared to 31% for SubTrack-Grad. This demonstrates a 49% wall-time reduction compared to GaLore. Additionally, the memory required for storing optimizer states in SubTrack-Grad, using the last gradient matrix, is equivalent to GaLore's, as confirmed experimentally in Figures 3a, 3b, and 3c.

**Reducing Optimization Jumps.** We argue that GaLore, which relies on a single gradient matrix at each subspace update step for adjustment via SVD, is susceptible to noise from data and less important features. Figure 2 compares SubTrack-Grad and GaLore on the Ackley function to illustrate potential jumps and deviations in the optimization process caused by GaLore's projection approach. These jumps prevent GaLore from reaching the global minimum within 100 steps when the scale factor is set to 1. While increasing the scale factor to 3 enables GaLore to reach the global minimum, it also results in larger jumps due to the higher scale factor. Additionally, Figure 5 in Appendix G illustrates that the overall convergence during both pre-training and fine-tuning is similar to other methods when using SubTrack-Grad.

Table 4: Optimizer's memory requirement in each method along with the time complexity of involved SVD operations, considering a gradient matrix of dimension $m \times n$ and projection rank $r$ where $r \ll m \leq n$.

|  | Optimizer Memory | SVD Time Complexity |
|---|---|---|
| **Adam** | $2mn$ | − |
| **GaLore** | $mr + 2nr$ | $O(nm^2)$ |
| **SubTrack(Acc)-Grad** | $mn + mr + 2nr$ | $O(mr^2)$ |
| **SubTrack(Last)-Grad** | $mr + 2nr$ | $O(mr^2)$ |

**Runtime Consistency.** Figure 4 compares the wall-times of GaLore and SubTrack-Grad across subspace update intervals ranging from 50 to 500 while fine-tuning RoBERTa-Base on the COLA task with an NVIDIA T4 GPU and pre-training a Llama-based architecture with 60M parameters on the C4 dataset using an NVIDIA A100 GPU. The subspace update interval denotes the number of iterations between two updates; thus, increasing this interval reduces the update frequency. As shown, GaLore's runtime significantly increases with more frequent subspace updates, whereas SubTrack-Grad maintains minimal runtime overhead regardless of the update frequency. In Table 14 different performance achieved on COLA dataset for different subspace update intervals can be compared, illustrating that increasing the update frequency significantly raises GaLore's runtime, while the performance of the two methods remains very similar.

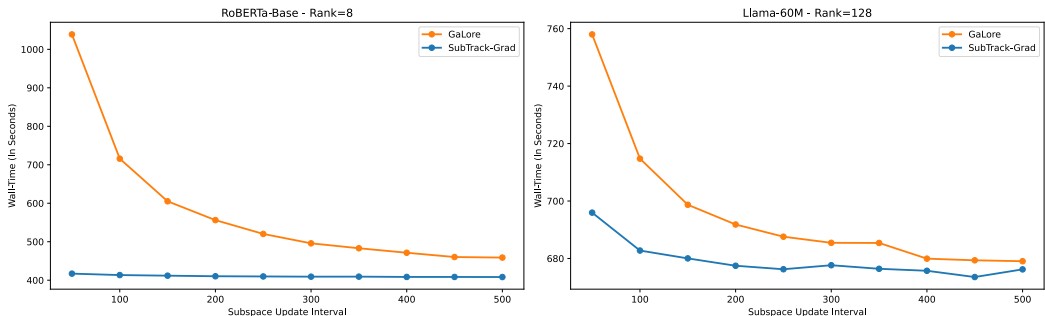

Figure 4: Comparing runtimes of GaLore and SubTrack-Grad. RoBERTa-Base is fine-tuned for 10 epochs on COLA and Llama-60M is pre-trained for 2500 iterations on C4 dataset. Subspace update intervals range from 50 to 500 in both experiments. Notice that by increasing subspace update interval, the update frequency actually decreases, as it indicates the number of iterations between two subspace update steps. As demonstrated, GaLore's wall-time can increase drastically when update frequency increases.

## 6 DISCUSSION AND CONCLUSION

We proposed a computationally efficient method that projects gradients into a lower-dimensional subspace, and preserves the previously computed subspace to incorporate gradient components from the orthogonal complement to perform rank-1 updates. This approach reduces the frequency of abrupt transitions between iterations and leverages the available information effectively.

In some cases, the extent of changes in the subspace may require updates of rank greater than 1; the pre-training of Llama-based architectures is a clear example of this need. During our experiments, we observed that applying updates as per equation 7, using the full-rank SVD of the tangent vector from equation 6, can hinder convergence if the singular values of the tangent vector become very small. Furthermore, simultaneously updating vector spaces associated with different singular values caused convergence issues in some cases. Therefore, we restricted updates to rank-1 in this paper, as this approach still enabled SubTrack-Grad to achieve performance comparable to or better than GaLore, with reduced runtime. In future work, we plan to explore increasing the rank of updates without compromising convergence. Additionally, dynamically selecting the step size could eliminate the need for manual tuning as a hyperparameter and further enhance convergence.

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

## A  CONVERGENCE OF SUBTRACK-GRAD

**Theorem 4.1 (Convergence of SubTrack-Grad)** *Suppose gradient has the following form (also equation 1) with functions $A_i$, $B_i$, and $C_i$ being L-continuous as per **Def. B.1** with constants $L_A$, $L_B$, and $L_C$ w.r.t. weight matrix $W_t$; and $\|W_t\|_F \leq M$; where $W_t$ denotes the weight matrix at step t, and M is a scalar value,*

$$G = \sum_i A_i + \sum_i B_i W C_i.$$

*Now, define $\widehat{B}_{i,t} = (S_{i,t}^l)^\top B_i(W_t) S_{i,t}^l$ and $\widehat{C}_{i,t} = (S_{i,t}^r)^\top C_i(W_t) S_{i,t}^r$, where $S_{i,t}^l$ and $S_{i,t}^r$ are the rank-r left and right projection matrices; $B_i(W_t)$ and $C_i(W_t)$ denote the dependence of $B_i$ and $C_i$ on the weight matrices $W_t$. Further letting $P_t = S_t^{l\top} G_t S_t^r$, and $\kappa_t = \frac{1}{N} \sum_i \lambda_{min}(\widehat{B}_{i,t})\lambda_{min}(\widehat{C}_{i,t})$, where $\lambda_{min}(\cdot)$ denotes the minimum eigenvalue over each batch, and N representing the number of samples in a batch. Assuming that the projection matrices remain constant during the training. Then for learning-rate $\mu$ and $min(\kappa_t) > (L_A + 2L_B L_C M^2)$, the SubTrack-Grad, with $\rho_t \equiv 1$ (the element-wise regularizer of the optimizer) satisfies:*

$$\|P_t\|_F \leq [1 - \mu(\kappa_{t-1} - L_A - 2L_B L_C M^2)]\|P_{t-1}\|_F.$$

*That is, $P_t \to 0$ and SubTrack-Grad converges.*

**proof.** To demonstrate that SubTrack-Grad converges to the global minimum during training, we begin by deriving the recursive form of the gradients.

Let $\otimes$ denote the Kronecker product. Then, $vec(AXB) = (B^\top \otimes A)vec(X)$.

By applying $vec$ to the gradient form given in the theorem, we obtain:

$$g_t = vec(G_t) = vec(\sum_i A_i + \sum_i B_i W C_i) = a_t - D_t w_t \tag{11}$$

where $g_t := vec(G_t)$, $w_t := vec(W_t)$, $a_t := \frac{1}{N} \sum_i vec(A_{i,t})$, and $D_t = \frac{1}{N} \sum_i C_{i,t} \otimes B_{i,t}$.

As defined in the theorem, let $P_t = S_t^{l\top} G_t S_t^r$. Its vectorized form can be expressed using the Kronecker product as follows:

$$
\begin{aligned}
p_t = vec(P_t) = vec(S_t^{l\top} G_t S_t^r) &= (S_t^{r\top} \otimes S_t^{l\top})vec(G_t) \\
&= (S_t^r \otimes S_t^l)^\top vec(G_t) = (S_t^r \otimes S_t^l)^\top g_t
\end{aligned} \tag{12}
$$

Now recalling $\widehat{G}_t$ from equation 10, it can be written as:

$$\widehat{G}_t = S_t^l S_t^{l\top} G_t S_t^r S_t^{r\top}$$

Thus, its vectorized form will be:

$$
\begin{aligned}
vec(\widehat{G}_t) = \widehat{g}_t = vec(S_t^l S_t^{l\top} G_t S_t^r S_t^{r\top}) &= vec(S_t^l P_t S_t^{r\top}) \\
&= (S_t^r \otimes S_t^l)vec(P_t) = (S_t^r \otimes S_t^l)p_t
\end{aligned} \tag{13}
$$

This is where the constant subspace assumption becomes necessary. To derive the recursive form of $g_t$, we assume that the projection matrices remain fixed throughout training, i.e., $S_t^r = S^r$ and $S_t^l = S^l$. Consequently, we can restate equations equation 12 and equation 13 as follows:

$$p_t = (S^r \otimes S^l)^\top g_t \tag{14}$$

$$\widehat{g}_t = (S^r \otimes S^l)p_t \tag{15}$$

Then we can write the recursive form of $g_t$:

$$
\begin{aligned}
g_t = a_t - D_t w_t &= (a_t - a_{t-1}) + (D_{t-1} - D_t)w_t + a_{t-1} - D_{t-1}w_t \\
&= e_t + a_{t-1} - D_{t-1}(w_{t-1} + \mu\widehat{g}_{t-1}) = e_t + g_{t-1} - \mu D_{t-1}\widehat{g}_{t-1}
\end{aligned} \tag{16}
$$

where $e_t := (a_t - a_{t-1}) + (D_{t-1} - D_t)w_t$.

Note that in deriving equation 16, we utilized the general form of the weight update rule, $w_{t+1} = w_t - \mu g_t$, which can be rewritten as $w_t = w_{t+1} + \mu g_t$. By applying this rule along with equation 11, we arrive at the second equality in equation 16 as follows:

$$g_t = a_t - D_t w_t = a_t - D_t w_t - g_{t-1} + g_{t-1}$$
$$= a_t - D_t w_t - a_{t-1} + D_{t-1} w_{t-1} + a_{t-1} - D_{t-1} w_{t-1}$$
$$= a_t - D_t w_t - a_{t-1} + D_{t-1}(w_t + \mu g_{t-1}) + a_{t-1} - D_{t-1}(w_t + \mu g_{t-1})$$
$$= a_t - D_t w_t - a_{t-1} + D_{t-1} w_t + \mu D_{t-1} g_{t-1} + a_{t-1} - D_{t-1} w_t - \mu D_{t-1} g_{t-1}$$
$$= a_t - a_{t-1} + (D_{t-1} - D_t) w_t + a_{t-1} - D_{t-1}$$

To obtain $p_t$ from this recursive formulation, we can left-multiply by $\left(S^r \otimes S^l\right)^\top$, as shown in equation 15:

$$p_t = \left(S^r \otimes S^l\right)^\top e_t + \left(S^r \otimes S^l\right)^\top g_{t-1} - \mu \left(S^r \otimes S^l\right)^\top D_{t-1} \widehat{g}_{t-1} \tag{17}$$

Now, based on equation 14 and equation 15, $p_t$ can be written as:

$$p_t = \left(S^r \otimes S^l\right)^\top e_t + p_{t-1} - \mu \left(S^r \otimes S^l\right)^\top D_{t-1} (S^r \otimes S^l) p_{t-1} \tag{18}$$

Let define:

$$\widehat{D}_t := \left(S^r \otimes S^l\right)^\top D_t (S^r \otimes S^l) = \frac{1}{N} \sum_i \left(S^r \otimes S^l\right)^\top (C_{i,t} \otimes B_{i,t})(S^r \otimes S^l)$$
$$= \frac{1}{N} \sum_i (S^{r\top} C_{i,t} S^r) \otimes (S^{l\top} B_{i,t} S^l) \tag{19}$$

Then we can expand equation 18 and show that:

$$p_t = (I - \mu \hat{D}_{t-1}) p_{t-1} + \left(S^r \otimes S^l\right)^\top e_t \tag{20}$$

Note that $S^l$ and $S^r$ are orthonormal matrices. This is ensured because the subspace is initialized using the SVD of $G_0$, and the Grassmannian update rule provided in equation 7 preserves the orthonormality of the subspace matrices throughout training. Since $S^l$ and $S^r$ are orthonormal, we have $S^{l\top} S^l = I$ and $S^{r\top} S^r = I$. Consequently, we can bound the norm of the second term in equation 20 as follows:

$$\|\left(S^r \otimes S^l\right)^\top e_t\|_2 = \|vec(S^{l\top} E_t S^r)\|_2 = \|S^{l\top} E_t S^r\|_F \leq \|E_t\|_F \tag{21}$$

Here $E_t$ is the matrix form of $e_t$, and as declared before, $e_t := (a_t - a_{t-1}) + (D_{t-1} - D_t) w_t$, thus:

$$E_t := \frac{1}{N} \sum_i (A_{i,t} - A_{i,t-1}) + \frac{1}{N} \sum_i (B_{i,t-1} W_t C_{i,t-1} - B_{i,t} W_t C_{i,t}) \tag{22}$$

Next, we need to find an upper bound for the norm of each term in equation 22 to establish an upper bound for $\|E_t\|_F$. Based on the assumptions of the theorem, $A_i$, $B_i$, and $C_i$ exhibit L-Lipschitz continuity with constants $L_A$, $L_B$, and $L_C$, respectively. Additionally, $\|W_t\|_F$ is bounded by a scalar $M$. We have:

$$\|A_t - A_{t-1}\|_F \leq L_A \|W_t - W_{t-1}\|_F = \mu L_A \|\tilde{G}_{t-1}\|_F \leq \mu L_A \|P_{t-1}\|_F \tag{23}$$

In the first equality, we apply equation 8, while the last equality holds due to equation 15 and the orthonormality of the projection matrices. The subsequent two inequalities can be derived similarly using these equations.

$$\|(B_t - B_{t-1}) W_t C_{t-1}\|_F \leq L_B \|W_t - W_{t-1}\|_F \|W_t\|_F \|C_{t-1}\|_F$$
$$= \mu L_B L_C M^2 \|P_{t-1}\|_F \tag{24}$$

$$\|B_t W_t (C_{t-1} - C_t)\|_F \leq L_C \|B_t\|_F \|W_t\|_F \|W_{t-1} - W_t\|_F$$
$$= \mu L_B L_C M^2 \|P_{t-1}\|_F \tag{25}$$

We can now derive the bound for $\|E_t\|_F$ as follows:

$$\|E_t\|_F \leq \mu L_A \|\tilde{G}_{t-1}\|_F \leq \mu L_A \|P_{t-1}\|_F + \mu L_B L_C M^2 \|P_{t-1}\|_F + \mu L_B L_C M^2 \|P_{t-1}\|_F$$
$$= \mu(L_A + 2L_B L_C M^2)\|P_{t-1}\|_F \quad (26)$$

To calculate the norm bound for the first term in equation 20, we first need to establish the bounds for $\widehat{D}_t$. This involves estimating the minimum eigenvalue of $\widehat{D}_t$.

If we define $\gamma_{min,i,t} = \lambda_{min}(S^{l\top} B_{i,t} S^l) \lambda_{min}(S^{r\top} C_{i,t} S^r)$, then it follows that $\lambda_{min}((S^{l\top} B_{i,t} S^l) \otimes (S^{r\top} C_{i,t} S^r)) = \gamma_{min,i,t}$. Consequently, $\widehat{D}_t$ will satisfy the following inequality for every unit vector $\boldsymbol{v}$:

$$\boldsymbol{v}^\top \widehat{D}_t \boldsymbol{v} = \frac{1}{N} \sum_i \boldsymbol{v}^\top \left[ (S^{l\top} B_{i,t} S^l) \otimes (S^{r\top} C_{i,t} S^r) \right] \boldsymbol{v} \geq \frac{1}{N} \sum_i \gamma_{min,i,t} \quad (27)$$

this actually provides a lower bound for eigenvalues of $\widehat{D}_t$, thus:

$$\lambda_{\max}(I - \mu \widehat{D}_{t-1}) \leq 1 - \frac{\mu}{N} \sum_i \gamma_{min,i,t-1} \quad (28)$$

considering the definition of $\kappa_t$ in the theorem, we can now easily show that:

$$\|P_t\|_F \leq [1 - \mu(\kappa_{t-1} - L_A - 2L_B L_C M^2)]\|P_{t-1}\|_F.$$

and completing the proof.

While SubTrack-Grad utilizes right/left projections to reduce memory consumption, the proof is presented using both projection matrices to ensure generality. Here, we demonstrate how the proof proceeds under the assumption $m \leq n$ (without loss of generality), which allows the use of the left projection matrix.

Using the left projection matrix, the current formulation of $P_t$, defined as $P_t = S_t^{l\top} G_t S_t^r$, simplifies to $P_t = S_t^{l\top} G_t$. Similarly, $\widehat{G}_t = S_t^l S_t^{l\top} G_t S_t^r S_t^{r\top}$ reduces to $\widehat{G}_t = S_t^l S_t^{l\top} G_t$. From this point, the proof continues by substituting $S_t^r$ with the identity matrix, allowing the derivation of the vectorized forms of $g_t$, $\widehat{g}_t$, $p_t$, and related terms.

The remainder of the proof remains largely unaffected. It can be readily verified that the recursive formulation of $g_t$ is unchanged. Although the definition of $P_t$ is modified, it continues to satisfy the bounds required for convergence, ensuring that $P_t$ converges to 0 when the left projection matrix is used.

## B   GRASSMANN EXPONENTIAL

**Theorem 4.2 (Grassmann Exponential)** *Let $P = UU^T \in Gr(n,p)$ be a point on the Grassmannian manifold, where $U \in St(n,p)$ is the orthonormal basis of the corresponding subspace. Consider a tangent vector $\Delta \in T_P Gr(n,p)$, and let $\Delta_U^{hor}$ denote the horizontal lift of $\Delta$ to the horizontal space at $U$ in the Stiefel manifold $St(n,p)$. Suppose the thin SVD of $\Delta_U^{hor}$ is given by $\Delta_U^{hor} = \hat{Q} \Sigma V^T$, where $\hat{Q} \in St(n,r)$, $\Sigma = diag(\sigma_1, \ldots, \sigma_r)$ contains the nonzero singular values of $\Delta_U^{hor}$ with $r = \min(p, n-p)$, and $V \in St(p,r)$. The Grassmann exponential map, representing the geodesic emanating from $P$ in the direction $\Delta$, is given by:*

$$Exp_P^{Gr}(t\Delta) = \left[ UV \cos(t\Sigma)V^T + \hat{Q} \sin(t\Sigma)V^T + UV_\perp V_\perp^T \right],$$

*where $V_\perp \in \mathbb{R}^{p \times (p-r)}$ is any orthogonal complement of $V$.*

**proof.** Using Grassmannina mathematics, we know that every $\Delta \in T_P Gr(n,p)$ is of the form

$$\Delta = Q \begin{pmatrix} 0 & B^T \\ B & 0 \end{pmatrix} Q^T = \left[ Q \begin{pmatrix} 0 & -B^T \\ B & 0 \end{pmatrix} Q^T, P \right] \quad (29)$$

Then the lift of $\Delta \in T_P Gr(n, p)$ to $Q = (U \quad U_\perp)$ can also be calculated explicitly as follows:

$$\Delta_Q^{\text{hor}} = [\Delta, P]Q = Q \begin{pmatrix} 0 & -B^T \\ B & 0 \end{pmatrix} \tag{30}$$

To resume our proof, we need to define the orthogonal group and specifying its tangent space.

**Definition B.1 ( Orthogonal Group)** *The orthogonal group $O(n)$ is defined as the set of all $n \times n$ matrices $Q$ over $\mathbb{R}$ such that $Q^T Q = QQ^T = I_n$, where $Q^T$ is the transpose of $Q$ and $I_n$ is the $n \times n$ identity matrix:*

$$O(n) = \{Q \in \mathbb{R}^{n \times n} \mid Q^T Q = I_n = QQ^T\}.$$

Then the tangent space of the orthogonal group $O(n)$ at a point $Q$, denoted $T_Q O(n)$, is defined as the set of matrices of the form $Q\Omega$, where $\Omega \in \mathbb{R}^{n \times n}$ is a skew-symmetric matrix, i.e., $\Omega^T = -\Omega$:

$$T_Q O(n) = \{Q\Omega \mid \Omega \in \mathbb{R}^{n \times n}, \Omega^T = -\Omega\}.$$

The geodesic from $Q \in O(n)$ in direction $Q\Omega \in T_Q O(n)$ is calculated via

$$\text{Exp}_Q^O(tQ\Omega) = Q \exp_m(t\Omega), \tag{31}$$

If $P \in Gr(n, p)$ and $\Delta \in T_P Gr(n, p)$ with $\Delta_Q^{\text{hor}} = Q \begin{pmatrix} 0 & -B^T \\ B & 0 \end{pmatrix}$, the geodesic in the Grassmannian is therefore

$$\text{Exp}_P^{Gr}(t\Delta) = \pi^{OG} \left( Q \exp_m \left( t \begin{pmatrix} 0 & -B^T \\ B & 0 \end{pmatrix} \right) \right). \tag{32}$$

where $\pi^{OG}$ is the projection from $O(n)$ to $Gr(n, p)$. If the thin SVD of $B$ is given by

$$B = U_\perp^T \Delta_U^{\text{hor}} = U_\perp^T \hat{Q} \Sigma V^T$$

with $W := U_\perp^T \hat{Q} \in St(n - p, r), \Sigma \in \mathbb{R}^{r \times r}, V \in St(p, r)$. Let $W_\perp, V_\perp$ be suitable orthogonal completions. Then,

$$\exp_m \begin{pmatrix} 0 & -B^T \\ B & 0 \end{pmatrix} = \begin{pmatrix} V & V_\perp & 0 & 0 \\ 0 & 0 & W & W_\perp \end{pmatrix} \begin{pmatrix} \cos(\Sigma) & 0 & -\sin(\Sigma) & 0 \\ 0 & I_{p-r} & 0 & 0 \\ \sin(\Sigma) & 0 & \cos(\Sigma) & 0 \\ 0 & 0 & 0 & I_{n-p-r} \end{pmatrix} \begin{pmatrix} V^T & 0 \\ V_\perp^T & 0 \\ 0 & W^T \\ 0 & W_\perp^T \end{pmatrix},$$

which leads to the desired result when inserted into equation 32. For more mathematical details, you can refer to Edelman et al. (1998), Bendokat et al. (2024), or other useful resources on Grassmann geometry.

## C  MEMORY AND TIME COMPARISON

Table 5 presents the wall-times measured to compare the computational efficiency of SubTrack-Grad and GaLore. Additionally, Table 6 shows the peak memory consumption for each architecture during training, using the hyperparameters detailed in 13, on an NVIDIA A100 GPU with 40GB of memory. These values were used to create the bar plots shown in 3a and 3d. For wall-time comparisons, each architecture was trained for 2,000 steps, ensuring exactly 10 subspace updates while excluding evaluation steps. The number of subspace updates was doubled compared to the fine-tuning experiments because larger models and more complex tasks naturally require more frequent updates. This adjustment allowed for more accurate measurements considering this demand. For memory comparisons, a batch size of 8 was used across all architectures to maintain almost consistent memory requirements for the input data.

Table 7 and Table 8 present the wall-time and peak memory consumption, respectively, for fine-tuning RoBERTa-Base on GLUE tasks. These values were used to generate the bar plots shown in 3c and 3f. For wall-time comparisons, the same settings were used, but evaluation steps were excluded, and fine-tuning was limited to 2500 iterations to ensure exactly 5 subspace update steps during the process.

Table 5: Wall-time comparison for pre-training Llama-based architectures with varying model sizes on the C4 dataset. The experiments consist of 2000 iterations, corresponding to exactly 10 subspace update steps. The last two columns present the average percentage change in runtime relative to Full-Rank and GaLore, respectively.

| | 60M rank=128 | 130M rank=256 | 350M rank=256 | 1B rank=512 | 3B rank=512 | w.r.t FR | w.r.t GaLore |
|---|---|---|---|---|---|---|---|
| **Full-Rank** | 524.0 | 1035.1 | 1396.4 | 974.9 | 1055.9 | - | - |
| **GaLore** | 547.8 | 1094.2 | 1589.0 | 1729.5 | 2715.5 | +53.94% | - |
| **SubTrack-Grad** | 534.9 | 1061.0 | 1465.9 | 1191.1 | 1385.7 | +13.08% | -26.54% |

Table 6: Peak memory consumption comparison when pre-training Llama-based architectures with different sizes on C4 dataset. The last two columns present the average percentage change in memory consumption relative to Full-Rank and GaLore, respectively.

| | 60M rank=128 | 130M rank=256 | 350M rank=256 | 1B rank=512 | 3B rank=512 | w.r.t FR | w.r.t GaLore |
|---|---|---|---|---|---|---|---|
| **Full-Rank** | 2.47 | 3.67 | 7.24 | 18.48 | 34.57 | - | - |
| **GaLore** | 2.36 | 3.47 | 6.42 | 14.88 | 25.81 | -20.32% | - |
| **SubTrack-Grad** | 2.39 | 3.48 | 6.55 | 15.39 | 26.76 | -17.91% | +3.02% |

Table 9 and Table 10 present the wall-time and peak memory consumption for each method when fine-tuning RoBERTa-Large on SuperGLUE tasks. These values were used to generate the bar plots shown in 3b and 3e. For wall-time comparisons, as with GLUE tasks, evaluation steps were omitted to ensure accurate measurements of fine-tuning time, and the number of iterations was adjusted to include exactly 5 subspace updates during fine-tuning.

## D  FINE-TUNING ROBERTA

RoBERTa-Base was fine-tuned on GLUE tasks using the hyperparameters detailed in Table 11, matching those reported in the GaLore (Zhao et al., 2024) for rank-$4$ and rank-$8$ subspaces, with a subspace update interval set at 500 iterations.

We also fine-tuned RoBERTa-Large on SuperGLUE tasks using the hyperparameters from Luo et al. (2024), as detailed in Table 12, with the exception that we fine-tuned each task for 30 epochs.

## E  PRE-TRAINING LLAMA-BASED ARCHITECTURES

To manage resources, we pre-trained all five Llama-based architectures for 10,000 iterations using hyperparameters reported in Table 13. While larger models typically require more iterations, this setup was sufficient for comparing methods rather than creating perfectly optimized models.

## F  TIME AND PERFORMANCE CONSISTENCY

Table 14 demonstrates that increasing the update frequency significantly increases GaLore's runtime, while the performance of both methods remains comparable. This underscores SubTrack-Grad's efficiency in reducing runtime without sacrificing performance. Notably, this advantage becomes even more pronounced for tasks that demand more frequent updates.

Table 7: Wall-time comparison when fine-tuning RoBERTa-Base on GLUE tasks for different ranks $r$ up to a number of iterations to exactly include 5 subspace update steps. The last two columns present the average percentage change in runtime relative to Full-Rank and GaLore, respectively.

| | COLA | STS-B | MRPC | RTE | SST-2 | MNLI | QNLI | QQP | w.r.t FR | w.r.t GaLore |
|---|---|---|---|---|---|---|---|---|---|---|
| **Full-Rank** | 114.5 | 131.5 | 146.4 | 279.0 | 107.1 | 152.3 | 167.4 | 126.8 | - | - |
| **GaLore** (r=4) | 195.7 | 195.1 | 200.2 | 325.8 | 185.7 | 206.8 | 216.5 | 190.4 | +40.20% | - |
| **SubTrack(Acc)-Grad** (r=4) | 155.7 | 158.0 | 172.7 | 305.7 | 147.5 | 175.0 | 192.2 | 155.8 | +19.40% | -14.78% |
| **SubTrack(Last)-Grad** (r=4) | 152.6 | 161.1 | 170.2 | 303.8 | 155.0 | 178.5 | 191.8 | 157.3 | +20.05% | -14.31% |
| **GaLore** (r=8) | 188.0 | 195.6 | 196.5 | 328.3 | 187.1 | 208.0 | 217.1 | 189.4 | +39.58% | - |
| **SubTrack(Acc)-Grad** (r=8) | 149.4 | 159.1 | 171.2 | 304.9 | 148.9 | 177.1 | 192.1 | 156.8 | +19.14% | -14.65% |
| **SubTrack(Last)-Grad** (r=8) | 151.2 | 160.8 | 170.6 | 305.1 | 151.0 | 177.1 | 193.0 | 158.9 | +19.86% | -14.13% |

Table 8: Peak memory consumption when fine-tuning RoBERTa-Base on GLUE tasks for different ranks $r$. The last two columns present the average percentage change in memory consumption relative to Full-Rank and GaLore, respectively.

| | COLA | STS-B | MRPC | RTE | SST-2 | MNLI | QNLI | QQP | w.r.t FR | w.r.t GaLore |
|---|---|---|---|---|---|---|---|---|---|---|
| **Full-Rank** | 3.55 | 4.05 | 3.85 | 13.39 | 3.42 | 21.97 | 14.92 | 11.08 | - | - |
| **GaLore** (r=4) | 3.23 | 3.93 | 3.52 | 12.45 | 3.68 | 21.73 | 14.59 | 10.26 | -3.78% | - |
| **SubTrack(Acc)-Grad** (r=4) | 3.45 | 4.17 | 3.90 | 12.31 | 3.17 | 19.85 | 14.29 | 10.14 | -6.51% | -2.84% |
| **SubTrack(Last)-Grad** (r=4) | 3.23 | 3.62 | 3.52 | 12.45 | 2.93 | 21.84 | 13.86 | 9.93 | -6.40% | -2.73% |
| **GaLore** (r=8) | 3.23 | 3.93 | 3.53 | 12.45 | 3.68 | 21.74 | 14.59 | 10.26 | -3.67% | - |
| **SubTrack(Acc)-Grad** (r=8) | 3.45 | 4.18 | 3.90 | 12.32 | 3.18 | 19.86 | 14.29 | 10.15 | -6.40% | -2.83% |
| **SubTrack(Last)-Grad** (r=8) | 3.23 | 3.62 | 3.53 | 12.45 | 2.93 | 21.84 | 13.86 | 9.94 | -6.40% | -2.83% |

# G  CONVERGENCE VS. WALL-TIME

Figure 5 depicts the changes in the training loss function relative to wall-time. The results demonstrate that SubTrack-Grad's wall-time reduction has minimal impact on the learning process, showcasing a convergence pattern comparable to other methods without introducing significant computational overhead.

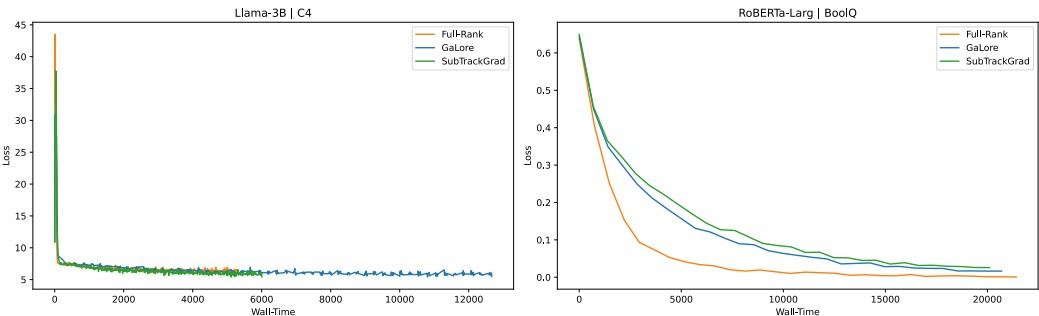

Figure 5: The figures present changes in training loss relative to wall-time for pre-training the Llama-3B architecture on the C4 dataset and fine-tuning RoBERTa-Large on the BoolQ dataset. They demonstrate that SubTrack maintains the overall learning process for both pre-training and fine-tuning while avoiding significant computational overhead.

Table 9: Wall-time comparison when fine-tuning RoBERTa-Large on SuperGLUEup to a number of iterations to exactly include 5 subspace update steps. The last two columns present the average percentage change in runtime relative to Full-Rank and GaLore, respectively.

| | BoolQ | CB | COPA | WIC | WSC | $AX_b$ | $AX_g$ | w.r.t FR | w.r.t GaLore |
|---|---|---|---|---|---|---|---|---|---|
| **Full-Rank** | 1406.3 | 171.5 | 35.1 | 250.4 | 216.4 | 470.7 | 52.1 | - | - |
| **GaLore** (r=8) | 1536.8 | 272.2 | 235.3 | 483.0 | 393.2 | 661.4 | 219.9 | +46.07% | - |
| **SubTrack-Grad** (r=8) | 1468.0 | 198.1 | 82.2 | 326.1 | 266.7 | 542.0 | 86.6 | +14.09% | -21.89% |

Table 10: Peak memory consumption comparison when fine-tuning RoBERTa-Large on SuperGLUE. The last two columns present the average percentage change in memory consumption relative to Full-Rank and GaLore, respectively.

| | BoolQ | CB | COPA | WIC | WSC | $AX_b$ | $AX_g$ | w.r.t FR | w.r.t GaLore |
|---|---|---|---|---|---|---|---|---|---|
| **Full-Rank** | 20.16 | 15.15 | 7.86 | 7.96 | 9.11 | 11.51 | 7.81 | - | - |
| **GaLore** (r=8) | 19.08 | 12.53 | 5.32 | 5.94 | 7.26 | 7.98 | 5.34 | -20.32% | - |
| **SubTrack-Grad** (r=8) | 19.09 | 12.40 | 5.12 | 5.95 | 7.06 | 8.00 | 5.35 | -20.84% | -0.66% |

Table 11: Hyperparameters of fine-tuning RoBERTa-Base on GLUE tasks.

| | MNLI | SST-2 | MRPC | CoLA | QNLI | QQP | RTE | STS-B |
|---|---|---|---|---|---|---|---|---|
| Batch Size | 16 | 16 | 16 | 32 | 16 | 16 | 16 | 16 |
| # Epochs | 30 | 30 | 30 | 30 | 30 | 30 | 30 | 30 |
| Learning Rate | 1E-05 | 1E-05 | 3E-05 | 3E-05 | 1E-05 | 1E-05 | 1E-05 | 1E-05 |
| SubTrack(Acc)-Grad Step-Size | 0.001 | 0.001 | 1.5 | 0.1 | 0.0001 | 0.001 | 1.0 | 1.0 |
| SubTrack(Last)-Grad Step-Size | 0.1 | 0.001 | 5.0 | 5.0 | 0.01 | 1.0 | 8.0 | 10.0 |
| Rank Config. | | | | $r=4$ | | | | |
| $\alpha$ | | | | 4 | | | | |
| Max Seq. Len. | | | | 512 | | | | |

| | MNLI | SST-2 | MRPC | CoLA | QNLI | QQP | RTE | STS-B |
|---|---|---|---|---|---|---|---|---|
| Batch Size | 16 | 16 | 16 | 32 | 16 | 16 | 16 | 16 |
| # Epochs | 30 | 30 | 30 | 30 | 30 | 30 | 30 | 30 |
| Learning Rate | 1E-05 | 2E-05 | 2E-05 | 1E-05 | 1E-05 | 2E-05 | 2E-05 | 3E-05 |
| SubTrack(Acc)-Grad Step-Size | 0.001 | 0.01 | 15.0 | 3.0 | 0.001 | 0.001 | 1.0 | 1.0 |
| SubTrack(Last)-Grad Step-Size | 0.1 | 0.1 | 5.0 | 13.0 | 0.1 | 1.0 | 5.0 | 10.0 |
| Rank Config. | | | | $r=8$ | | | | |
| $\alpha$ | | | | 2 | | | | |
| Max Seq. Len. | | | | 512 | | | | |

Table 12: Hyperparameters of fine-tuning RoBERTa-Large on SuperGLUE tasks.

| | BoolQ | CB | COPA | WIC | WSC | $AX_b$ | $AX_g$ |
|---|---|---|---|---|---|---|---|
| Batch Size | 16 | 16 | 16 | 16 | 16 | 16 | 16 |
| # Epochs | 30 | 30 | 30 | 30 | 30 | 30 | 30 |
| Learning Rate | 1e-5 | 1e-5 | 1e-5 | 1e-5 | 1e-5 | 1e-5 | 1e-5 |
| SubTrack-Grad Step-Size | 0.1 | 1.0 | 50.0 | 50.0 | 1 | 1.0 | 1.0 |
| Subspace Update Interval | 500 | 100 | 100 | 500 | 250 | 500 | 100 |
| Rank Config. | | | | $r=8$ | | | |
| $\alpha$ | | | | 4 | | | |
| Max Seq. Len. | | | | 512 | | | |

Table 13: Hyperparameters of pre-training Llama-based architectures.

|              | 60M  | 130M | 350M | 1B   | 3B    |
|--------------|------|------|------|------|-------|
| Hidden       | 512  | 768  | 1024 | 2048 | 4096  |
| Intermediate | 1376 | 2048 | 2736 | 5461 | 11008 |
| Heads        | 8    | 12   | 16   | 24   | 32    |
| Layers       | 8    | 12   | 24   | 32   | 32    |
| Batch Size   | 256  | 128  | 64   | 8    | 8     |
| Rank         | 128  | 256  | 256  | 512  | 512   |
| Steps        | 10K  | 10K  | 10K  | 10K  | 10K   |

Table 14: Comparing changes in performance while decreasing the subspace update interval (increasing the frequency of subspace updates) on fine-tuning RoBERTa-Base on COLA task.

|                              | 500    | 450    | 400    | 350    | 300    | 250     | 200    | 150    | 100    | 50      |
|------------------------------|--------|--------|--------|--------|--------|---------|--------|--------|--------|---------|
| GaLore's Wall-Time           | 458.79 | 460.19 | 471.36 | 483.04 | 495.83 | 520.28  | 556.14 | 605.1  | 715.87 | 1038.79 |
| GaLore's Performance         | 0.5358 | 0.5285 | 0.5385 | 0.5356 | 0.5309 | 0.54821 | 0.5332 | 0.5387 | 0.5463 | 0.5463  |
| SubTrack-Grad's Wall-Time    | 408.22 | 408.44 | 408.47 | 409.25 | 409.17 | 409.76  | 410.29 | 411.77 | 413.35 | 417.07  |
| SubTrack-Grad's Performance  | 0.5332 | 0.5382 | 0.5364 | 0.5385 | 0.5354 | 0.5364  | 0.5416 | 0.5385 | 0.5364 | 0.5673  |

