# OpenReview forum: "SubTrack your Grad: Gradient Subspace Tracking for Memory-Efficient LLM Training and Fine-Tuning"
_ICLR.cc/2025/Conference — Submitted to ICLR 2025_

### Official Review · Reviewer_nJ4a · 2024-10-30

**Soundness:** 3
**Presentation:** 2
**Contribution:** 3
**Rating:** 5
**Confidence:** 4

**Summary:**

This paper proposes a method, SubTrack-Grad, to update gradient subspaces using rank-1 estimation of tangent vector on the Grassmannian manifold. SubTrack-Grad reduces disruptive changes in the gradient update process, leading to more stable learning dynamics. Experimental results show that SubTrack-Grad accelerates convergence and achieves superior performance compared to GaLore.

**Strengths:**

1. SubTrack-Grad improves GaLore by introducing a method for updating gradients in a low-dimensional space. It addresses the issue that not all layer gradients evolve within stable low-rank subspaces, thus accelerating convergence through rank-1 estimation of tangent vectors on the Grassmannian manifold.
2. Experimental results show that SubTrack-Grad outperforms GaLore in both wall-time efficiency and accuracy, demonstrating faster convergence and improved performance.

**Weaknesses:**

1. The method rank-1 gradient update needs further explanations about why choosing rank-1 rather than rank-r, where r can be any integer, and more supporting related works. Providing empirical comparisons with higher rank updates, or discussing potential tradeoffs between update rank and performance/efficiency can make this point valid. Additionally, examples of related work that use rank-1 updates in similar contexts would strengthen this update.
2. SubTrack claims to reduce the memory usage compared to GaLore, but still periodically (at each update interval $k$) utilizes SVD to obtain the largest singular value and the corresponding singular vectors, which is also used in GaLore. Rather than inferring from the algorithmic description, specific memory usage measurements or GPU footprint comparisons between SubTrack-Grad and GaLore would provide more concrete evidence for or against the memory efficiency claims.
3. Experimental results are insufficient to fully evaluate SubTrack-Grad’s performance. Additional experiments are needed to provide a more comprehensive assessment (please see Questions for further details).
4. Some figures require clearer presentation and explanation, for example, Figure 1 lacks legends and uses an unclear line to describe the tangent vector, and Figure 2 needs more explanation for the right part.

**Questions:**

1. In the introduction section, the authors proposes that SubTrack-Grad is more memory-efficient than GaLore, and in the experiments, the authors evaluates the wall-time of both methods, but can you please provide the memory usage or GPU footprint to compare these two methods? Since in SubTrack-Grad, the authors utilizes SVD when computing the rank-1 estimation of tangent vector, while GaLore also utilizes SVD in their gradient projection. Can authors please explain more about the memory saving in the SubTrack-Grad, compared to GaLore?
2. SubTrack-Grad utilizes rank-1 update to reduce the abrupt changes in the optimization process and the performance and wall-time is better than GaLore, but can authors explain and give more explanation about why choosing rank-1 rather than rank-2 or rank-3? Or can authors provide more related works which utilize rank-1 update to support the method of SubTrack-Grad?
3. For the subspace update interval $k$ in the SubTrack-Grad, the authors compare the wall-time with different intervals, but can authors provide the comparison of the performance (accuracy) affected by the different update intervals? I think this can be better to help understand the influence of this factor.

---

> ### Author Response · Authors · 2024-11-23
>
> Thank you sincerely for your valuable review. We greatly appreciate your feedback. Below, we provide our responses to the concerns you raised:
>
> **1- Elaboration on Employing Rank-1 Updates**
>
> Thank you for your suggestion. We are deeply glad you paid close attention to the details of our method. In fact, we conducted some internal experiments to explore the effects of increasing the rank of updates. However, these experiments revealed additional complexities that require further investigation and careful implementation design, as mentioned in our **Discussion and Conclusion** section.
>
> SubTrack-Grad demonstrates its efficiency and effectiveness even when restricted to rank-1 updates. Given its promising results, we were eager to publish our work to share it with the community sooner rather than later, leaving this intriguing aspect for future research. Additionally, to the best of our knowledge, SubTrack-Grad is the first method to leverage subspace tracking for reducing memory footprint.
>
> **2- Memory and Computational Efficiency of SubTrack-Grad Along with Additional Experiments**
>
> We sincerely apologize for any confusion our statements may have caused. To clarify, SubTrack-Grad does not claim to reduce memory usage compared to GaLore; rather, it emphasizes significant improvements in **computational efficiency**.
>
> After expanding our experiments, we observed that eliminating the gradient accumulation step and using the last gradient preserves performance while reducing memory consumption. Further details on this modification can be found on line 215 of the rebuttal revision. We have included a **Time and Space Complexity** section (line 467), to better document the method’s complexities, as the following table summarizes.
>
> Table 4: Optimizer's memory requirement in each method along with the time complexity of involved SVD operations, considering a gradient matrix of dimension $m \times n$ and projection rank $r$ where $r \ll m \le n$.
>
> | Method                    | Optimizer Memory      | SVD Time Complexity   |
> |---------------------------|:-----------------------:|:------------------------:|
> | **Adam**                  | 2mn             | -                     |
> | **GaLore**                |  mr + 2nr        |  O(nm^2)          |
> | **SubTrack(Acc)-Grad**    |  mn + mr + 2nr   |  O(mr^2)          |
> | **SubTrack(Last)-Grad**   |  mr + 2nr        |  O(mr^2)          |
>
> It illustrate that the main power of our method lies in performing all additional computations in a low-dimensional space.
>
> We now present results for fine-tuning RoBERTa-Large on SuperGLUE tasks, while we have extended pretraining experiments with Llama models, scaling up to 3B parameters. The results highlight that SubTrack-Grad reduces wall-time by up to **20.57% on GLUE tasks (15% average reduction)** and by **up to 65% on SuperGLUE tasks (22% average reduction)**. For a 3B parameter model, GaLore incurred a 157% increase in wall-time compared to full-rank training, whereas SubTrack-Grad exhibited only a 31% increase, representing a **49% reduction in wall-time compared to GaLore**. This improvement comes with only a minor increase in memory consumption—at most 3.7% on the 3B model.  For further details, please refer to the **Experiments and Results** section of the rebuttal revision.
>
> **3- Comparing Performance Changes with Increased Subspace Update Frequency**
>
> We are happy to provide this information. **Table 14** in **Appendix F** of our rebuttal revision demonstrates that increasing the subspace update frequency significantly increases GaLore's runtime, while the performance of both methods remains comparable. This underscores SubTrack-Grad's efficiency in reducing runtime without compromising performance.
>
> **4- Adjustments to Figures**
>
> Thank you for your feedback. We have made the necessary adjustments, which are now reflected in our rebuttal revision.
>
> We sincerely appreciate your constructive feedback and would be more than happy to provide any further clarification.

---

> ### Comment · Reviewer_nJ4a · 2024-11-27
>
> Thank you for your response. I’m still unclear about the main differences between rank-1 and rank-$r$ updates. Additionally, in Table 14, there appears to be a clear relationship between update frequency and performance. Are these results robust enough to support such a conclusion (linear relationship)? Can authors provide some explanations or references?

---

> > ### Author Response · Authors · 2024-11-28
> >
> > Thank you for your response. We are happy to provide further clarification.
> >
> > **Rank-1 vs. Rank-r Updates**
> >
> > Our internal experiments revealed that higher-rank updates can hinder convergence due to issues associated with the small singular values of the SVD of the tangent vector. Specifically:
> > - *Noise Sensitivity*: Small singular values are more susceptible to noise, as SVD is inherently sensitive to data perturbations.
> > - *Numerical Instability*: These small values can cause numerical instability, potentially leading to arithmetic errors.
> > - *Convergence Challenges*: The combined effects of noise sensitivity and numerical instability can disrupt the convergence process and alter training dynamics.
> >
> > To address these challenges, various subspace tracking strategies have been proposed, such as using cutoff thresholds, dynamically adjusting update ranks, and more. However, identifying the optimal approach for tracking the **gradient** subspace requires further experimentation and analysis.
> >
> > Moreover, there is extensive literature on robust subspace tracking that focuses on isolating the primary subspace from noise to mitigate such issues (e.g., [1], [2]). In light of these considerations, rank-1 updates offer a simple yet effective solution for our current problem, yielding promising results. Nevertheless, we plan to investigate employing higher-rank updates in future research.
> >
> > **Linear Correlation Between Subspace Update Frequency and Performance**
> >
> > The aim of the table is not to suggest a direct linear correlation. As discussed in [3], different layers exhibit varying behaviors: some stabilize early during training, while others continue to fluctuate between subspaces. These dynamics are influenced by factors such as the data, pre-trained parameters, and network structure, and this single experiment is not enough for getting to a correlation conclusion.
> >
> > In Table 14, we demonstrate that SubTrack-Grad achieves performance comparable to or better than GaLore, without incurring significant computational overhead. Thus, **in scenarios requiring higher update frequencies**, SubTrack-Grad proves to be an efficient and effective method.
> >
> > We hope we have addressed your concerns. Please feel free to let us know if anything remains unclear or needs further clarification. We greatly appreciate and look forward to engaging in constructive discussions.
> >
> > ---
> >
> > [1] Namrata Vaswani, Thierry Bouwmans, Sajid Javed, and Praneeth Narayanamurthy. Robust subspace learning: Robust PCA, robust subspace tracking, and robust subspace recovery. *IEEE Signal Processing Magazine*, 35(4):32–55, July 2018. ISSN 1558-0792. doi: 10.1109/msp.2018.2826566.
> > [2] Nguyen Viet Dung, Nguyen Linh Trung, Karim Abed-Meraim, et al. Robust subspace tracking algorithms in signal processing: A brief survey. *REV Journal on Electronics and Communications*, 11(1-2), 2021.
> > [3] Ajay Jaiswal, Lu Yin, Zhenyu Zhang, Shiwei Liu, Jiawei Zhao, Yuandong Tian, and Zhangyang Wang. From GaLore to WeLore: How low-rank weights non-uniformly emerge from low-rank gradients, 2024.

---

> > ### Author Response · Authors · 2024-12-01
> >
> > Dear Reviewer,
> >
> > As we approach the end of the discussion period, we wanted to reach out to ensure that our responses have adequately addressed the concerns you raised. We greatly appreciate your valuable feedback and have incorporated your suggestions into the revised rebuttal. We are eager to hear your thoughts on the changes and whether you believe they have improved the quality of the paper. Please let us know if there are any remaining issues or additional clarifications you require from us.
> >
> > Thank you for your time and consideration.

---

### Official Review · Reviewer_EzYJ · 2024-11-03

**Soundness:** 3
**Presentation:** 3
**Contribution:** 2
**Rating:** 5
**Confidence:** 4

**Summary:**

Galore's original paper employs SVD to compute the projection matrix. To enhance the efficiency of this calculation, the paper introduces Gradient Subspace Tracking (SubTrack-Grad), which optimizes within a compact core subspace of the gradient matrices and dynamically tracks changes using Grassmannian manifold geometry. Experiments test the effectiveness of SubTrack-Grad.

**Strengths:**

- The paper clearly presents its ideas, and the proposed SubTrack-Grad is straightforward to understand.

- The paper analyzes the convergence of the proposed SubTrack-Grad.

-  Experiments demonstrate that the proposed SubTrack-Grad's performance is comparable to Galore's, with improvements in computational efficiency.

**Weaknesses:**

- When compared to Galore using SVD, the actual reduction in running time is very small, and there appears to be no advantage in running time when compared to Galore using QR decomposition. The Galore compared in the paper only uses a projection matrix, and it is not necessary to use SVD, but only QR decomposition, which is more efficient.

- The paper lacks experimental comparisons with Galore on memory usage, i.e., the peak memory consumption. The reduction of running time only makes sense if the memory increase is not large, otherwise why not use a larger batch size. It is better to test the memory usages when different model sizes are used.

- The proposed SubTrack-Grad does not show a significant advantage over Galore. The paper needs to compare memory usage, running time, and performance at the same time. The reduction in running time is meaningful only when performance and memory usage are both comparable to Galore.

**Questions:**

- Galore can compute the projection matrix P using QR decomposition, which is significantly more efficient. The paper needs a thorough comparison with Galore using QR decomposition for projection matrix calculation.

- Memory usage should be compared with Galore. Galore requires only a low-rank projection matrix, while the proposed SubTrack-Grad necessitates storing two matrices, Gacc and the projection matrix S.

- When reporting running time, it is better to present the total fine-tuning or training time for the same number of iterations, allowing readers to accurately assess the speedup ratio over Galore.

- Since Galore is the main compared method, it is a reasonable choice to strictly follow Galore's experimental setups.

---

> ### Author Response · Authors · 2024-11-23
>
> Thank you sincerely for your valuable review. We greatly appreciate your feedback. Below, we provide our responses to the concerns you raised:
>
> **1- Peak Memory Consumption and Our Advantage Over GaLore**
>
> We sincerely appreciate the reviewer’s feedback. We are now reporting memory consumption alongside wall-time reductions for all our models and datasets to provide further support for our claims. We have included results for fine-tuning RoBERTa-Large on SuperGLUE tasks, in addition to our earlier experiments. We have also extended our experiments with Llama models, scaling up to 3B parameters.
> Notably, after expanding our experiments, we observed that eliminating the gradient accumulation step and using the last gradient preserves performance while reducing memory consumption with results presented in **Table 1**. Further details on this modification are included on line 215 of the rebuttal version. We have included a **Time and Space Complexity** section (line 467), to better document the method’s complexities, as the following table summarizes.
>
> Table 4: Optimizer's memory requirement in each method along with the time complexity of involved SVD operations, considering a gradient matrix of dimension $m \times n$ and projection rank $r$ where $r \ll m \le n$.
>
> | Method                    | Optimizer Memory      | SVD Time Complexity   |
> |---------------------------|:-----------------------:|:------------------------:|
> | **Adam**                  | 2mn             | -                     |
> | **GaLore**                |  mr + 2nr        |  O(nm^2)          |
> | **SubTrack(Acc)-Grad**    |  mn + mr + 2nr   |  O(mr^2)          |
> | **SubTrack(Last)-Grad**   |  mr + 2nr        |  O(mr^2)          |
>
> It illustrate that the main power of our method lies in performing all additional computations in a low-dimensional space.
>
> We also doubled the number of subspace update steps included in wall-time measurement experiments on Llama models, to ensure a more robust and valid measurement. The results demonstrate that SubTrack-Grad reduces wall-time by up to **20.57% on GLUE tasks (15% average reduction)** and by **up to 65% on SuperGLUE tasks (22% average reduction)**, with the **same** memory-consumption as GaLore’s. For a 3B parameter model, GaLore incurred a substantial 157% increase in wall-time compared to full-rank training, whereas SubTrack-Grad exhibited only a 31% increase, representing a **49% reduction in wall-time (26% average reduction) compared to GaLore**, while showing minor memory consumption increase, **at most 3.7%** on the 3B model.  **Figure 3** will give a better comparison via visualization for all datasets. Also, all supporting tables are included in **Appendix C**.
>
> **2- Using the Same Number of Iterations for Fair Comparison**
>
> The reported runtime for different methods has been measured using consistent hyperparameters. However, we understand that presenting performance and runtime data in the same table may have caused some confusion, as the number of iterations used to report performance differs from those used for wall-time experiments.
>
> In response to your feedback, we have separated these reports to avoid further misunderstandings. Performance results are presented in **Tables 1, 2, and 3**, while **Figure 3** provides a summary of memory and wall-time comparisons. Detailed tables are included in **Appendix C** of the rebuttal revision. We hope this revision clarifies any confusion.
>
> **3- Using GaLore’s Experimental Setups**
>
> Thank you for your feedback. We confirm that we used the exact same settings as those reported in GaLore’s experiments. This can be verified by comparing the values provided in the **Appendix D** and **Appendix E** sections.
>
>
> **4- Using QR Decomposition to Improve the Efficiency of GaLore**
>
> Thank you for this suggestion. We are open to exploring the integration of QR decomposition with GaLore, however, we believe this change may not be straightforward in this state, given GaLore's original formulation. Their method and theorems indicate that to minimize performance loss caused by projecting the gradient into a lower-dimensional space, the gradient matrix must be projected onto the eigenvectors corresponding to the largest eigenvalues. To address this, a possible approach is to use SVD, which leverages singular values to project gradients onto the subspace that retains the most critical information.
>
> While QR decomposition is computationally more efficient than SVD, it may lead to information loss and, consequently, performance degradation. Since our current focus is on maintaining or improving GaLore’s performance while enhancing efficiency, we are cautious about adopting QR decomposition at this stage. We would really appreciate your further feedbacks regarding this integration.
>
> We sincerely appreciate your constructive feedback and would be more than happy to provide any further clarification.

---

### Official Review · Reviewer_wAms · 2024-11-03

**Soundness:** 2
**Presentation:** 2
**Contribution:** 2
**Rating:** 6
**Confidence:** 3

**Summary:**

The paper introduces SubTrack-Grad, a novel method for memory-efficient training and fine-tuning of large language models (LLMs). This technique focuses on confining the optimization process to a core subspace of the gradient matrices and tracking its evolution dynamically using Grassmannian manifold geometry. The key improvements claimed over existing methods, like GaLore, include computational efficiency and better runtime with minimal accuracy trade-offs. The authors demonstrate that SubTrack-Grad achieves comparable or superior performance while reducing computational costs by up to 20% in some cases.

**Strengths:**

The introduction of the SubTrack-Grad method, which leverages Grassmannian manifold geometry and rank-1 updates, presents an innovative approach to tracking gradient subspaces efficiently. This approach reduces computational overhead compared to periodic SVD in similar methods.
 The experimental results achieves comparable or superior performance to existing methods like GaLore.
The paper includes a theoretical analysis with convergence guarantees.

**Weaknesses:**

The derivation of Equation 7 appears disconnected from the preceding content. Include a reference to the original source or derivation details. Additionally, clarify the rationale for applying cosine and sine functions to the singular values in the projection update.

I can't confirm whether \( S_{t+1} \) maintains orthonormality post-update, as it is essential for the method's validity.

The justification for reduced computational load is unclear. Specifically, computing \( G_t \) (the full gradient) and updating \( S_{t-1} \) to project to \( A \) may introduce substantial overhead. This undermines claims of computational benefits. Strengthen arguments or provide concrete runtime comparisons.

The element-wise regularizer mentioned in Section 4 is undefined in the pseudocode and algorithmic description.

The notations \( N \), \( L_A \), \( L_B \), and \( L_C \) in Theorem 4.1 require explicit definitions for clarity.

The Appendix proof claiming convergence to the global minimum during training is questionable. Provide stronger theoretical backing or clarify the assumptions that make this feasible.

Theorem 4.1 suggests projecting the gradient twice, differing from the algorithm's single-projection description. Ensure alignment between theoretical claims and practical implementation.

The proof assumes fixed projection matrices throughout training, contradicting the stated goal of dynamically tracking subspace changes. Address this inconsistency.

The paper refers to \( w \) as components from the gradient decomposition rather than model parameters. Clarify why \( w_{t+1} = w_t - \alpha \nabla \) is used in Equation 16 to avoid confusion.

**Questions:**

The paper has some technical issues which sounds not tangible and correct. Please see weakness.

---

> ### Author Response · Authors · 2024-11-23
>
> Thank you sincerely for your valuable review. We appreciate your feedback and are honored by your kind description of our method as novel! Below, we provide our responses to the concerns you raised:
>
> **1- Subspace Update Rule Presented in Equation 7**
>
> As this formulation directly applies Grassmannian geometry, we initially considered that providing detailed explanations might exceed the scope of our paper. However, we recognize the value of further clarification.
>
> The update rule ensures that movement follows the Grassmannian geodesic. Specifically, starting from $S_t$, it allows us to reach $S_{t+1}$ with a step size of $\eta$, while ensuring that $S_{t+1}$ remains on the Grassmannian manifold, thereby preserving all the essential properties required for our algorithm. The inclusion of sine and cosine functions naturally arises from the mathematical framework of traversing the Grassmannian manifold.
>
> We have added the main theorem and relevant definitions to the **Theoretical Analysis** section of the rebuttal revision, starting line 331. Furthermore, the detailed proof has been included in **Appendix B** for additional clarity.
>
> **2- Exploring Computational Efficiency**
>
> The reviewer is absolutely right. In response, we have now reported memory consumption alongside wall-time reductions for all our models and datasets. It is worth noting that, after expanding our experiments, we observed that eliminating the gradient accumulation step and using the last gradient maintains performance while reducing memory consumption, and thus we expanded our experiment using that advantage. Further details on this modification are provided on line 214 of the rebuttal version, with performance comparisons reported in **Table 1**. Additionally, we have included a **Time and Space Complexity** section (line 467), to better document the method’s complexities, as the following table summarizes.
>
> Table 4: Optimizer's memory requirement in each method along with the time complexity of involved SVD operations, considering a gradient matrix of dimension $m \times n$ and projection rank $r$ where $r \ll m \le n$.
>
> | Method                    | Optimizer Memory      | SVD Time Complexity   |
> |---------------------------|:-----------------------:|:------------------------:|
> | **Adam**                  | 2mn             | -                     |
> | **GaLore**                |  mr + 2nr        |  O(nm^2)          |
> | **SubTrack(Acc)-Grad**    |  mn + mr + 2nr   |  O(mr^2)          |
> | **SubTrack(Last)-Grad**   |  mr + 2nr        |  O(mr^2)          |
>
> It illustrate that the main power of our method lies in performing all additional computations in a low-dimensional space.
>
> Our results demonstrate that SubTrack-Grad reduces wall-time by up to **20.57% on GLUE tasks (15% average reduction)** and by **up to 65% on SuperGLUE tasks (22% average reduction)**. Notably, for a 3B parameter model, GaLore incurred a substantial 157% increase in wall-time compared to full-rank training, whereas SubTrack-Grad exhibited only a 31% increase, representing a **49% reduction in wall-time (26% in average) compared to GaLore**. For further details, please refer to the **Experiments and Results** section of the rebuttal revision.
>
> **3- Element-Wise Regularizer Not Included in the Method**
>
> The element-wise regularizer is related to the optimizer, which determines each parameter's learning rate and state. These parameters are used for projecting the gradients back during backpropagation and updating the model parameters, and it does not influence the gradient projection or subspace updates directly. To address any potential confusion, we have included additional details on line 206 of the rebuttal revision.
>
> **4- Specifying Notations in Theorem 4.1**
>
> We have updated the theorem to include all the notations, which can be found on line 308 of the rebuttal revision.
>
> **5- Convergence to the Global Minimum**
>
> We apologize if the proof has caused confusion. While we assert that SubTrack-Grad can converge to a minimum, we do not claim that this minimum is guaranteed to be the global minimum. This is a common limitation of many gradient-based optimization methods, including GaLore. Achieving convergence to the global minimum typically requires additional techniques or assumptions, which are beyond the scope of our current work.
>
> **6- Inconsistency Between Theory and Practical Implementation**
>
> Your observation is absolutely valid. In the original proof, we utilized full projection matrices to ensure generalizability. To provide consistency, we have now included the necessary adjustments to adapt the proof for the use of only left/right projection matrix, following the very same procedure. These updates can be found on line 780 in **Appendix A**, demonstrating that the proof seamlessly generalizes to align with our practical implementation.

---

> > ### Author Response · Authors · 2024-11-23
> >
> > **7- Assumption of Fixed Projection Matrices**
> >
> > The assumption of fixed projection matrices was utilized to derive the recursive formulation in Equation 16, and a similar assumption was made in the proof of convergence for GaLore. This assumption does not hinder the validity of the proof, as the frequency of gradient subspace updates remains low.
> >
> > Moreover, SubTrack-Grad offers an advantage over GaLore that further supports this assumption. Unlike GaLore, which performs SVD on the new gradient matrix—resulting in a rank-$r$ change in the subspace and increased vulnerability to noise—SubTrack-Grad employs only rank-1 updates. This approach ensures that the updated subspace closely resembles the previous one.
> >
> > By maintaining a low update frequency and carefully controlling the magnitude of changes during subspace updates, we ensure that the theoretical proof is fully consistent with the practical implementation of SubTrack-Grad.
> >
> > **8- Clarifying $ w $ Notation in the Paper**
> >
> > In our paper, $ W_t $ refers to the weight matrix of the network at time $ t $, while $ w_t $ represents the vectorized form of this weight matrix. We hope this clarification resolves any ambiguity, and we try to maintain consistency throughout the paper's notations.
> >
> > **9- Clarifying the Process in Equation 16**
> >
> > We have now clarified how this update rule, which in our paper's notation is expressed as $ w_{t+1} = w_t - \mu g_t $, was applied in deriving Equation 16. The explanation has been added on line 702 of our rebuttal revision.
> >
> > We sincerely appreciate your constructive feedback and would be more than happy to provide any further clarification.

---

> ### Author Response · Authors · 2024-12-01
>
> Dear Reviewer,
>
> As we approach the end of the discussion period, we wanted to reach out to ensure that our responses have adequately addressed the concerns you raised. We greatly appreciate your valuable feedback and have incorporated your suggestions into the revised rebuttal. We are eager to hear your thoughts on the changes and whether you believe they have improved the quality of the paper. Please let us know if there are any remaining issues or additional clarifications you require from us.
>
> Thank you for your time and consideration.

---

### Official Review · Reviewer_51cV · 2024-11-04

**Soundness:** 2
**Presentation:** 2
**Contribution:** 2
**Rating:** 3
**Confidence:** 3

**Summary:**

The paper introduces SubTrack-Grad, a method to make LLM training more memory and time-efficient. Methods like GaLore reduce memory by projecting gradients into low-rank spaces using SVD. But SVD is computationally expensive. So, instead SubTrack-Grad updates a compact gradient subspace with lightweight, rank-1 adjustments on the Grassmannian manifold.

**Strengths:**

- SubTrack-Grad  shows reduction in memory footprint and computational cost of training LLMs . This method also reports 15-20% reduction in runtime is substantial and valuable for scaling model training.
- The use of Grassmannian manifold geometry to track gradient subspaces is innovative. This allows the method to dynamically adapt to shifts in gradient directions avoiding frequent SVD operations.
- By controlling subspace adjustments through rank-1 updates, this method reduces abrupt changes and noise in the optimization process.

**Weaknesses:**

The paper in its current form has several gaps in fully addressing and validating the main claims. For instance, the main benefit of the method, which is to improve memory efficiency isn't validated in the experiments.  Could you report the peak memory usage for  SubTrack-Grad and other baselines along with perplexity and time in the experiments?

 The benefits behind the gradient accumulation that SubTrack-Grad proposes isn't intuitively clear from the paper. The experiments are a bit vague and limited on this. In fact, the experiments in general are quite limited. The paper could benefit from experiments on "larger" models >3b parameters.

Minor: Some other related works that can be cited:

1. Adam-mini: Use Fewer Learning Rates To Gain More, Zhang et al
2. BlockLLM: Memory-Efficient Adaptation of LLMs by Selecting and Optimizing the Right Coordinate Blocks: Ramesh et al

**Questions:**

Refer to weaknesses section

---

> ### Author Response · Authors · 2024-11-23
>
> Thank you sincerely for your valuable review. We appreciate your feedback and are honored by your kind description of our method as innovative! Below, we provide our responses to the concerns you raised:
>
> **1- Benefits of Gradient Accumulation**
>
> We initially used gradient accumulation to reduce noise in gradient matrices across different samples. However, during our exploration of ways to improve the efficiency of our method, we discovered that performing subspace tracking using the latest calculated gradient matrix delivers comparable performance while being more memory-efficient. Based on this, we extended our experiments to incorporate this minor modification. The performance comparison presented in **Table 1** demonstrates that even with the latest gradient matrix, SubTrack-Grad outperforms or matches the results of GaLore. Further details on this modification can be found on line 214 of the rebuttal version.
>
> **2- Lack of Experiments and the Need to Report Memory Consumption**
>
> The reviewer is absolutely right. We are pleased to report that we have significantly expanded our experiments. Results for fine-tuning RoBERTa-Large on SuperGLUE tasks, in addition to our earlier experiments are now included. Meanwhile, we extended experiments with Llama models, scaling up to 3B parameters. We made sure that memory consumption alongside wall-time reductions are reported for all of our models and datasets.
>
> We also doubled the number of subspace update steps included in wall-time measurement experiments on Llama models, to ensure a more robust and valid measurement. The results demonstrate that SubTrack-Grad reduces wall-time by up to **20.57% on GLUE tasks (15% average reduction)** and by **up to 65% on SuperGLUE tasks (22% average reduction)**, with the same memory-consumption as GaLore’s. Notably, for a 3B parameter model, GaLore incurred a substantial 157% increase in wall-time compared to full-rank training, whereas SubTrack-Grad exhibited only a 31% increase, representing a **49% reduction in wall-time compared to GaLore**, while showing only a **3.7% increase** memory overhead.
> We have included the tables reporting memory and wall-time on pre-training Llama architectures for your reference. **Figure 3** will give a better comparison via visualization for all datasets. Also, all supporting tables are included in **Appendix C**.
>
> ***
>
> Table 5: Wall-time comparison for pre-training Llama-based architectures with varying model sizes on the C4 dataset. The experiments consist of 2000 iterations, corresponding to exactly 10 subspace update steps. The last two columns present the **average** percentage change in runtime relative to Full-Rank and GaLore, respectively.
>
> | **Model Size**      | **60M (rank=128)** | **130M (rank=256)** | **350M (rank=256)** | **1B (rank=512)** | **3B (rank=512)** | **w.r.t FR** | **w.r.t GaLore** |
> |----------------------|:--------------------:|:---------------------:|:---------------------:|:-------------------:|:-------------------:|:--------------:|:------------------:|
> | **Full-Rank**        | 524.0             | 1035.1              | 1396.4              | 974.9            | 1055.9           | -            | -                |
> | **GaLore**           | 547.8             | 1094.2              | 1589.0              | 1729.5           | 2715.5           | +53.94%      |  -               |
> | **SubTrack-Grad**    | 534.9             | 1061.0              | 1465.9              | 1191.1           | 1385.7           | +13.08%      | **-26.54%**          |
>
> ***
>
> Table 6: Peak memory consumption comparison when pre-training Llama-based architectures with different sizes on C4 dataset. The last two columns present the **average** percentage change in memory consumption relative to Full-Rank and GaLore, respectively.
>
> | Method          | 60M rank=128 | 130M rank=256 | 350M rank=256 | 1B rank=512 | 3B rank=512 | w.r.t FR      | w.r.t GaLore |
> |------------------|:--------------:|:---------------:|:---------------:|:-------------:|:-------------:|:-------------------:|:--------------:|
> | **Full-Rank**    | 2.47         | 3.67          | 7.24          | 18.48       | 34.57       | -             | -            |
> | **GaLore**       | 2.36         | 3.47          | 6.42          | 14.88       | 25.81       | -20.32%       | -            |
> | **SubTrack-Grad**| 2.39         | 3.48          | 6.55          | 15.39       | 26.76       | -17.91%       | **+3.02%**       |
>
> ***
>
> **3- Adding Citations**
>
> Thank you for highlighting these valuable works. We have incorporated both references into the paper, and you can now find them in lines 124 and 137 of the revised rebuttal version.
>
> We sincerely appreciate your constructive feedback and would be more than happy to provide any further clarification.

---

> > ### Comment · Reviewer_51cV · 2024-11-25
> > **Thank the authors for their response**
> >
> > I thank the authors for their response . Thanks for adding experiments on peak memory consumption. Subtrack-grad seems to perform better than full-rank. But as the model increases in size, it seems that it starts to consume more memory than GaLore. Are you able the analyze this behavior?

---

> > > ### Author Response · Authors · 2024-11-25
> > >
> > > Thank you for your attention. We are pleased to provide this information.
> > >
> > > The primary focus of SubTrack-Grad at this stage is to reduce the computational overhead caused by GaLore’s resource-intensive operations. Notably, the memory overhead of our approach for Llama models (ranging in size from 60M to 3B) is limited to **1.28% to 3.7%**, while the runtime reduction spans **2.35% to 49%**. This represents a substantial improvement in runtime performance with only a marginal increase in memory usage. Below, we outline the space complexity of both methods to substantiate our claims.
> > >
> > > Both SubTrack (using the last gradient matrix) and GaLore share the **same constant memory requirements**: $mr$ space for storing the projection matrix and $2nr$ for the optimizer state (see the Time and Space Complexity section, line 467, for more details).
> > > The distinction in peak memory usage between GaLore and SubTrack-Grad arises from the temporary memory needed to compute the new subspace. GaLore employs SVD for subspace calculation, requiring $m^2 + m + mn = O(mn)$ temporary memory. In contrast, SubTrack-Grad involves the following key steps:
> > >
> > > 1. Solving the least squares problem ($O(mn)$).
> > > 2. Computing the residual matrix ($mn$).
> > > 3. Calculating the tangent vector ($mr$).
> > > 4. Updating the subspace ($mr$).
> > >
> > > **Consequently, the temporary memory complexity for both methods is of the same order: $O(mn)$.**
> > >
> > > Moreover, since these steps are sequential, temporary memory usage can be further reduced in practice through **in-place implementations and efficient garbage collection mechanisms**, which we plan to integrate before releasing our code publicly.
> > >
> > > Please let us know if we could provide any further information.

---

> > > > ### Comment · Reviewer_51cV · 2024-11-29
> > > > **Response to feedback**
> > > >
> > > > Thank you for providing the complexity of the method. This clarifies the memory requirements of Subtrack-Grad. Given that the main objective is to reduce the computational cost and training time, this work has not compared with other methods which have much lower computational cost and time (eg. BlockLLM, BAdam, Adam-mini). In practice, GaLore is one of the methods which takes longer time to train.  Due to this, I would like to keep my score for now.

---

> > > > > ### Author Response · Authors · 2024-12-01
> > > > >
> > > > > Thank you so much for your response. Below, we would like to address some key points regarding your suggestions.
> > > > >
> > > > > SubTrack-Grad is designed to reduce runtime and memory consumption without compromising performance. While the suggested methods are all innovative and useful, we believe GaLore should be considered as our main baseline, and we have prioritized that to other baselines. Nevertheless we are happy to provide more-baselines in our camera-ready version, and here are the details about why we prioritized GaLore:
> > > > >
> > > > > **BlockLLM and BAdam:** Both BlockLLM and BAdam focus on selecting a subset of parameters and limiting optimization to those during each iteration. This approach is indeed effective for fine-tuning, offering time and memory efficiency. However, these methods have limitations for pre-training scenarios. When parameters are initialized randomly, freezing a significant portion of them often leads to performance degradation.
> > > > >
> > > > > For instance, BAdam does not compare its pre-training performance with other methods in its experiments; In Table 4 of BlockLLM, results show that as the model size increases, GaLore consistently outperforms BlockLLM, even though the experiments are limited to a 350M LLaMA model. In contrast, SubTrack-Grad optimizes all model parameters simultaneously without altering the learning dynamics, akin to GaLore. This is why we chose GaLore as our primary baseline, though we are open to incorporating comparisons with other methods in the camera-ready version.
> > > > >
> > > > > **Adam-Mini:** is orthogonal to our method and could be seamlessly combined with SubTrack-Grad or GaLore to further improve memory efficiency. We welcome the opportunity to explore and report the results of this combination in future iterations.
> > > > >
> > > > > **Advantages of SubTrack-Grad:** Tracking the subspace provides several advantages that can further enhance results. We leverage gradient information that methods like BlockLLM, BAdam, and GaLore do not utilize in optimization (by considering the estimation error which lies in the orthogonal subspace, we can leverage all available information, which can be insightful for dynamically adapt the training process too). This is a promising area for exploration, and we are excited to contribute these insights to the community to encourage further investigation.
> > > > >
> > > > > ---
> > > > >
> > > > > To the best of our knowledge, SubTrack-Grad is the first method to use subspace tracking to improving both memory and time efficiency. We hope you will reconsider your rating, as this work has the potential to establish an innovative framework that reduces resource requirements in a highly interpretable and insightful way, especially after the constructive discussion that we have with the reviewers. We tried to provide all necessary analyses addressing your concerns; please let us know if there is still any information that you may need.

---

> ### Author Response · Authors · 2024-11-27
>
> Dear Reviewer,
>
> Thank you once again for your thorough and thoughtful review. Your feedback has been invaluable in helping us strengthen our experiments.
>
> As we approach the conclusion of the discussion period, we wanted to ensure that we have addressed all of your concerns. If there are any remaining issues or additional information you require, please do not hesitate to let us know.
>
> We greatly value constructive discussions and would appreciate the opportunity to hear any further insights or suggestions you may have.

---

### Meta-Review · Area_Chair_fGSC · 2024-12-21

**Metareview:**

This paper introduces a new method for memory-efficient training and fine-tuning of large language models (LLMs). The method uses Grassmannian manifold geometry and rank-1 updates to track a compact gradient subspace, which helps reduce computational costs compared to methods like GaLore. SubTrack-Grad shows similar or better performance while cutting runtime by up to 20% in some cases. The theoretical part, including convergence guarantees, gives the work a strong foundation. The experiments show the method works well even for large models (up to 3B parameters), making it a useful approach for memory-efficient LLM training.
During the discussion, the authors addressed many issues raised by the reviewers. They added experiments, including memory usage comparisons, extended results for larger models, and gave more explanations about the theoretical parts, like rank-1 updates and subspace tracking. But even with these improvements, reviewers think the paper still needs more work before acceptance. For example, a deeper comparison with other baselines is needed, as well as a better analysis of the trade-offs in the method. The experimental setup could be expanded to evaluate memory, runtime, and performance more fully, and hyperparameters should be more consistent across tests. It would also help to make the figures and theoretical explanations clearer so readers can better understand the work.

**Additional Comments On Reviewer Discussion:**

The discussion among reviewers pointed out both the good points and areas where the paper can still improve. Reviewers appreciated that the authors responded well and added more experiments and explanations during the rebuttal. However, some important issues still need to be addressed. These include making stronger comparisons with other baselines, like GaLore with QR decomposition, and giving more detailed analysis of the trade-offs between rank-1 and higher-rank updates.

---

### Decision · Program_Chairs · 2025-01-22

Reject